# U-µP: THE UNIT-SCALED MAXIMAL UPDATE PARAMETRIZATION

**Charlie Blake**[*]
Graphcore

**Constantin Eichenberg**[*]
Aleph Alpha

**Josef Dean**
Graphcore

**Lukas Balles**
Aleph Alpha

**Luke Y. Prince**
Graphcore

**Björn Deiseroth**
Aleph Alpha

**Andres Felipe**[†]
**Cruz-Salinas**
Cohere

**Carlo Luschi**[‡]
Graphcore

**Samuel Weinbach**[‡]
Aleph Alpha

**Douglas Orr**
Graphcore

## ABSTRACT

The Maximal Update Parametrization (µP) aims to make the optimal hyperparameters (HPs) of a model independent of its size, allowing them to be swept using a cheap proxy model rather than the full-size target model. We present a new scheme, u-µP, which improves upon µP by combining it with Unit Scaling, a method for designing models that makes them easy to train in low-precision. The two techniques have a natural affinity: µP ensures that the scale of activations is independent of model size, and Unit Scaling ensures that activations, weights and gradients begin training with a scale of one. This synthesis opens the door to a simpler scheme, whose default values are near-optimal. This in turn facilitates a more efficient sweeping strategy, with u-µP models reaching a loss that is equal to or lower than comparable µP models and working out-of-the-box in FP8.

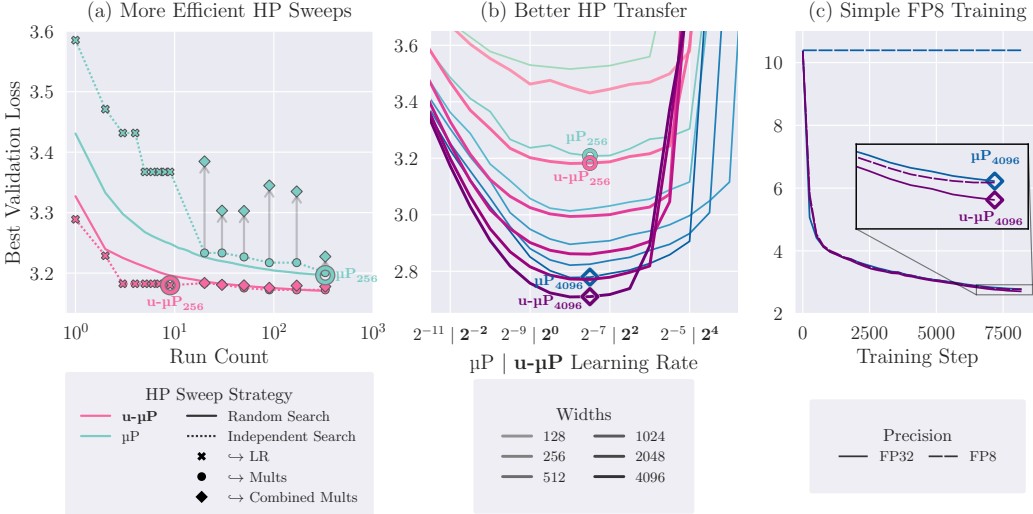

Figure 1: **(a)** Two different HP sweeping processes used for µP and u-µP proxy models. Unlike µP, u-µP admits independent (1D) search due to careful HP design. The first part of independent search is an LR sweep, which alone reaches near-optimal loss for u-µP. **(b)** Using the best proxy HPs from (a), we train many models at different widths and LRs. The best LR for width 256 is ~optimal for 4096, showing LR transfer along with lower loss. **(c)** We re-train with a simple un-scaled `.to(float8)` cast on matmul inputs. This would fail for other models, but u-µP trains with minimal degradation.

---

[*]Equal contribution.     [†]Work done while at Aleph Alpha.     [‡]Supervisory role.
Correspondence to: charlieb@graphcore.ai, constantin.eichenberg@aleph-alpha-ip.ai.

# 1 INTRODUCTION

The challenges of large-model training extend beyond the domain of engineering; they are also *algorithmic* in nature. Effective approaches for training smaller models are not guaranteed to work at the multi-billion-parameter scale used for today's large language models (LLMs). These difficulties can be framed in terms of stability, which we consider in three forms:

1. feature learning stability, which ensures that parts of the model do not learn too fast or slow relative to each other.
2. hyperparameter stability, which ensures that the optimal HPs for small models remain unchanged as the model size grows.
3. numerical stability, which ensures that floating-point representations during training stay within the range of a given number format.

The Maximal Update Parametrization (μP) (Yang & Hu, 2021; Yang et al., 2022) targets the first two sources of instability. μP defines a set of scaling rules that in principle make a model's optimal HP values consistent across model sizes and ensure 'maximal feature learning' in the infinite-width limit. The practical benefits of this are that models continue to improve as they get larger, and that practitioners can re-use a set of HP values (especially the learning rate) found for a small *proxy* version of their model, on a larger *target* model. This is vital for modern LLM training, where the cost of sweeping over candidate HP values for the target model is prohibitive. Consequently, μP has been adopted by several open LLM training efforts (Dey et al., 2023a;b; Liu et al., 2023; Hu et al., 2024) and there are indications of its use in state-of-the-art LLMs[1].

However, there exists a gap between the extensive theory underpinning μP and its effective use in practice. This relates to issues surrounding efficient HP search, HP transfer, interpretability, ease-of-use and low-precision training. Some of these problems have been observed in the literature (Yang et al., 2022; Almazrouei et al., 2023; Lingle, 2024); others we outline here for the first time. As a result, μP does not necessarily provide the kind of simple, stable scaling for which a user might hope.

To address this, we propose the Unit-Scaled Maximal Update Parametrization (u-μP). u-μP combines μP with another closely-related training innovation, Unit Scaling (Blake et al., 2023). μP ideally provides consistent training dynamics across model sizes, but says little about what those dynamics should be. Unit Scaling addresses this by proposing an ideal principle for dynamics: unit variance for all activations, weights and gradients. Unit Scaling was initially designed to ensure stable numerics, but in the context of μP the principle of unit-scale brings many additional benefits. We show that it provides a foundation upon which the broader range of drawbacks identified for μP can be addressed.

# 2 BACKGROUND

## 2.1 THE MAXIMAL UPDATE PARAMETRIZATION

Tensor Programs V (Yang et al., 2022) defines a parametrization as 'a rule for how to change [HPs] when the widths of a neural network change'. They show that μP is the only parametrization giving 'maximal feature learning' in the limit, while standard parametrization (SP) has imbalanced learning.

One consequence of this improved stability is that learning dynamics under μP are ideally independent of model-size, as are optimal HPs. This facilitates a method known as μTransfer, which describes the process of training many smaller proxy models to evaluate candidate HP values, then using the best-performing ones to train a larger target model.

**ABC-parametrizations**. μP, SP, and the Neural Tangent Kernel (NTK) (Jacot et al., 2018) are all instances of abc-parametrizations. This assumes a model under training where weights are defined as:

$$W_t = A_W \cdot w_t, \quad w_0 \sim \mathcal{N}(0, B_W^2),$$
$$w_{t+1} = w_t + C_W \cdot \Phi_t(\nabla\mathcal{L}_0, ..., \nabla\mathcal{L}_t), \tag{1}$$

---

[1] The GPT-4 technical report (OpenAI, 2023) hints at the use of μP by including Yang et al. (2022) in its references, without citing it directly. The multipliers present in the Grok (xAI, 2024) codebase also suggest the use of μP.

with $t$ a time-step and $\Phi_t(\nabla \mathcal{L}_0, ..., \nabla \mathcal{L}_t)$ is the weight update based on previous loss gradients.

A parametrization such as µP is then defined by specifying how scalars $A_W, B_W, C_W$ change with model width. This can be expressed in terms of width-dependent factors $a_W, b_W, c_W$, such that $A_W \propto a_W, B_W \propto b_W, C_W \propto c_W$. The values these factors take are what characterize a particular scheme. For µP these are given in Table 1. For depth, a similar result has been proved using depth-µP (Yang et al., 2023b), albeit in a restricted setting. When we refer to µP in the paper we assume the depth-µP scaling rules (Table 2, 'Residual' column).

A key property of the abc-parametrization is that one can shift scales between $A_W, B_W, C_W$ in a way that preserves learning dynamics (i.e. the activations computed during training are unchanged). We term this *abc-symmetry*. For a fixed $\theta > 0$, the behavior of a network trained with Adam is invariant to changes of the kind:

$$A_W \leftarrow A_W/\theta, \quad B_W \leftarrow B_W \cdot \theta, \quad C_W \leftarrow C_W \cdot \theta \qquad (2)$$

(reproduced from Tensor Programs V, Section J.2.1). This means that parametrizations like µP can be presented in different but equivalent ways. ABC-symmetry is a key component in developing u-µP.

**Transferable HPs**. µP focuses on the subset of HPs whose optimal values we expect to *transfer across* axes such as width and depth. We term these µTransferable HPs. All µTransferable HPs function as multipliers and can be split into three kinds: $\alpha_W, \sigma_W, \eta_W$ where $A_W \propto \alpha_W, B_W \propto \sigma_W, C_W \propto \eta_W$. The difference between these multipliers and the ones that define a parametrization is that they are specified by the user, rather than being a function of width. $\alpha_W$ and $\eta_W$ are rarely introduced outside of the µP literature, but can be valuable to tune for both µP and SP models. In the µP literature the term 'HPs' often implicitly refers to µTransferable HPs. We adopt this convention here, unless specified otherwise.

**Base shape**. Two additional non-µTransferable HPs introduced by µP are the `base-width` and `base-depth`. This refers to a mechanism where a user specifies a particular shape for the model, where its behavior under µP and SP are the same. The µP model still *scales* according to the abc-rules, so for all other shapes the two models will be different.

In summary, the absolute expressions for $A_W, B_W, C_W$ under µP are given by:

$$A_W \leftarrow \alpha_W \frac{a_W}{a_{W_{\text{base}}}}, \quad B_W \leftarrow \sigma_W \frac{b_W}{b_{W_{\text{base}}}}, \quad C_W \leftarrow \eta_W \frac{c_W}{c_{W_{\text{base}}}} \qquad (3)$$

Though base shapes are necessary for µP, they are not typically swept. Rather, they are considered a preference of the user, who may wish to retain the behavior of an existing SP model at a given shape.

**Choosing HPs to sweep**. In theory, the search space of µTransferable HPs includes $\alpha_W, \sigma_W, \eta_W$ for every parameter tensor $W$ in the model. In practice, far fewer HPs are swept, with global grouping often used for $\sigma_W$ and $\eta_W$, and many $\alpha_W$ are dropped or grouped across layers.

The sets of HPs chosen for sweeps in the µP literature is explored in Appendix E.1. Tensor Programs V uses a random search to identify the best HP values, which has become the standard approach to sweeping. The number of runs in a sweep is typically in the low 100s, incurring a non-negligible cost (though usually less than a single training run of the target model). This high number partly owes to dependencies between HPs (shown in Section 4.1), making the search space hard to explore.

Table 1: The scaling rules defining µP. The type of a weight is determined by whether fan-in & fan-out both depend on width (hidden), only fan-out does (input), or only fan-in (output). Hence fan-in is always a multiple of width here.

| | ABC-multiplier | | Input | Hidden | Output |
|---|---|---|---|---|---|
| | | | | Weight ($W$) Type | |
| | parameter | $(a_W)$ | 1 | 1 | $1/\text{fan-in}(W)$ |
| **µP** | initialization | $(b_W)$ | 1 | $1/\sqrt{\text{fan-in}(W)}$ | 1 |
| | Adam LR | $(c_W)$ | 1 | $1/\text{fan-in}(W)$ | 1 |

## 2.2 Low-precision training

All the major potential bottlenecks of model training—compute, communication and storage—see roughly linear improvements as the bit-width of their number format is reduced. In modern LLM training, the compute cost of large matrix multiplications (matmuls) means that substantial gains are available if these can be done in low-precision ($< 32$ bit) formats, which makes them one of the most promising avenues towards increased efficiency in deep learning.

Recent AI hardware offers substantial acceleration for the 8-bit FP8 E4 and E5 formats. However their reduced range means that they cannot directly represent some values generated during training. Various methods have been introduced to address this, such as the per-tensor dynamic re-scaling in Transformer Engine (NVIDIA, 2024b). However, this comes at the cost of added complexity and potential overheads. For a more in-depth treatment of low-precision formats, see Appendix J.

## 2.3 Unit Scaling

An alternative approach to low-precision training is Unit Scaling (Blake et al., 2023), which also uses per-tensor scaling factors to control range, but instead finds these factors via an analysis of expected tensor statistics at initialization. These are fixed factors, calculated independently of the contents of a tensor, at the beginning of training. As such, the method is easy to use and only adds the overhead of applying static scaling factors (which we show to be negligible in Appendix K).

These factors are chosen to ensure the unit variance of activations, weights and gradients at initialization. This is a useful criterion as it places values around the center of floating-point formats' absolute range. This applies to all tensors, meaning every operation in the network requires a scaling factor that ensures unit-scaled outputs, assuming unit-scaled inputs. Unit Scaling does not provide a mechanism for re-scaling tensors dynamically during training, but due to its ideal starting scale for gradients, activations and weights this may not be required. Empirically this is shown to be true across multiple architectures, though it is not guaranteed.

We provide an example of deriving the Unit Scaling rule for a matmul op in Appendix E.2, resulting in the scaling factor: $1/\sqrt{d_{\text{fan-in}}}$. We accompany this example with a full recipe for applying Unit Scaling to an arbitrary model.

## 3 The Unit-Scaled Maximal Update Parametrization

In this section we show how μP can be adapted to satisfy Unit Scaling, and provide a new set of HPs which—thanks to Unit Scaling—are more interpretable and separable than those commonly used for μP, unlocking several practical benefits. Although some features of u-μP presented here are transformer specific, we stress that u-μP can in principle be applied to a wide range of architectures, since both μP and Unit Scaling are very general approaches.

### 3.1 Combining μP with Unit Scaling

Whereas Unit Scaling provides rules for scaling all operations, μP only does so for parametrized ones. It's these operations we need to address to arrive at a unified scheme, resolving differences in the scaling rules each recommends. We begin with the expressions for the $A_W, B_W, C_W$ scaling factors in Equation (3), and substitute in the μP scaling rules defined in Table 1. This results in a full implementation of μP, which is shown in the top half of Table 2. We set out to turn this into a valid Unit Scaling scheme, which requires unit initializations ($B_W \leftarrow 1$) and matmuls with the Unit Scaling factor we identified in Section 2.3 ($A_W \leftarrow 1/\sqrt{\text{fan-in}}$).

Our first step is to drop the $\sigma_W$ and $\text{base-fan-in}$ HPs entirely, and associate the $\alpha_W$ HPs with subsequent non-linear functions instead of weights—decisions we justify in the rest of this section (this results in the simplified intermediate implementation in Table 11). Our input weights now have unit initializations as desired, and a unit parameter multiplier, which is also the appropriate scaling factor (as input layers here are embedding lookups, not matmuls).

Table 2: Scaling rules for µP versus u-µP, including associated HPs (assuming the *extended* set in Table 3). These rules constitute the definition of u-µP, along with the unit-scaled ops in Appendix B.

| | ABC-multiplier | | Input | Hidden | Output | Residual |
|---|---|---|---|---|---|---|
| | | | | Weight Type | | |
| **µP** | parameter | $(A_W)$ | $\alpha_{\text{emb}}$ | $1$ (or $\alpha_{\text{attn}}$) | $\alpha_{\text{out}}\frac{\text{base-fan-in}}{\text{fan-in}}$ | $\sqrt{\frac{\text{base-depth}}{\text{depth}}}$ * |
| | initialization | $(B_W)$ | $\sigma_{\text{init}}$ | $\sigma_{\text{init}}\sqrt{\frac{\text{base-fan-in}}{\text{fan-in}}}$ | $\sigma_{\text{init}}$ | — |
| | Adam LR | $(C_W)$ | $\eta\,\hat{\eta}_{\text{emb}}$ | $\eta\frac{\text{base-fan-in}}{\text{fan-in}}$ | $\eta$ | $\sqrt{\frac{\text{base-depth}}{\text{depth}}}$ |
| **u-µP** | parameter$^{\dagger}$ | $(A_W)$ | $1$ | $\frac{1}{\sqrt{\text{fan-in}}}$ | $\frac{1}{\text{fan-in}}$ ‡ | $\frac{1}{\sqrt{\text{depth}}}$ * |
| | initialization | $(B_W)$ | $1$ | $1$ | $1$ | — |
| | Adam LR | $(C_W)$ | $\eta\frac{1}{\sqrt{\text{fan-out}}}$ | $\eta\frac{1}{\sqrt{\text{fan-in}}}$ | $\eta$ | $\frac{1}{\sqrt{\text{depth}}}$ |

*Residual multipliers are applied to the end of each branch, rather than the output of linear layers.
†u-µP's $\alpha$ HPs are associated with operations, not weights, so are not included here (see Section 3.3).
‡To maintain unit scale we apply $1/\sqrt{\text{fan-out}}$ scaling in the backward pass (see Appendix H).

Hidden weights now have the implementation: $A_W \leftarrow 1$, $B_W \leftarrow \frac{1}{\sqrt{\text{fan-in}}}$, $C_W \leftarrow \eta\frac{1}{\text{fan-in}}$, which differs from our Unit Scaling criteria. However, using abc-symmetry (Equation (2)) we can shift scales by $\sqrt{\text{fan-in}}$, arriving at a unit-scaled scheme: $A_W \leftarrow \frac{1}{\sqrt{\text{fan-in}}}$, $B_W \leftarrow 1$, $C_W \leftarrow \eta\frac{1}{\sqrt{\text{fan-in}}}$.

Finally, our output layers also have unit initialization, but a parameter multiplier of $A_W \leftarrow 1/\text{fan-in}$. This differs from the Unit Scaling rule, but in the forward pass this is permissible as there are no subsequent matmuls in a transformer. In the backward pass this mis-scaling would propagate, so we apply the desired $\leftarrow 1/\sqrt{\text{fan-out}}$ factor. Using different forward and backward scales in this way is usually not allowed, but is valid for output layers due to the cut-edge rule (Appendix H).

The final change we make is to the input LR scaling rule, which we show in Section 3.4 is more effective if $c_W \leftarrow 1$ is replaced with $c_W \leftarrow 1/\sqrt{\text{fan-out}}$. With these changes made, we arrive at our final u-µP scheme, given in Table 2. Note that the scaling rules in this table must be combined with the standard Unit Scaling rules for other non-matmul operations. These are covered in Appendix B.

## 3.2 OUT-OF-THE-BOX LOW-PRECISION TRAINING

When training a transformer model with u-µP most tensors have stable scale during training, except a small number of *critical tensors* that exhibit scale growth, which can cause extreme values to go out of FP8 range. We empirically identify these to be the inputs to the attention dense projection and final FFN matmul as well as the weight of the decoder[2] (for details see Appendix A.7).

Based on our analysis, we propose the following FP8 mixed-precision scheme for u-µP transformers:

- For layers using non-critical tensors, we cast the input and weight to E4M3, and the gradient with respect to the output to E5M2. This is done in the forward computation, as well as the two backward computations (for the gradients w.r.t. the weight and input).
- For layers using critical tensors, matmuls are performed in BF16 (along with non-matmul operations). We keep optimizer states in FP32, leaving the use of FP8 to future work.

In some cases we are able to deal with the critical tensors by casting them to E5M2. However, we observed instabilities applying this in a large-scale setting. In small-scale settings we also empirically find that applying E4M3 instead of E5M2 for the gradients is possible, but again becomes problematic in the large-scale setting where gradients require higher dynamic range. Under our mixed-precision scheme, approximately 70% of the matmul computations in a Llama transformer block are performed in FP8. If desired, a dynamic per-tensor scaling could still be applied to the critical tensors.

---

[2] The decoder becomes negligible in terms of model flops as width and depth of the model increase, so we generally keep this operation in higher precision.

### 3.3 A PRINCIPLED APPROACH TO HYPERPARAMETERS

Approaches for selecting which HPs to sweep are poorly motivated in the literature (see Appendix C.2). Our objective in u-µP is to find a simple, well-justified and effective alternative. To this end, we propose the following ideal criteria:

1. **Minimal cardinality**: the use of as few HPs as possible.
2. **Maximal expressivity**: the ability to still express any model defined using the per-tensor $\alpha_W, \sigma_W, \eta_W$ HPs outlined in Section 2.1 (in practice, we relax this slightly).
3. **Minimal interdependency**: the optimal value of each HP should not depend on the value of other HPs, simplifying the search space.
4. **Interpretability**: there should be a clear explanation for what an HP's value 'means' in the context of the model.

The u-µP HPs given in Table 3 are designed to satisfy these criteria, to the fullest extent possible. The placement of these HPs in the model is given in Table 7.

**Cardinality & expressivity**. We arrive at our set of HPs in three steps, starting with the full $\alpha_W, \sigma_W, \eta_W$ for each weight tensor $W$. Firstly, we can remove any one of these HPs by permuting under abc-symmetry, such that one HP $= 1$. As we want our weights to begin with unit scale, we choose to drop $\sigma_W$. Secondly, we observe that several of the $\alpha_W$ HPs combine linearly with other $\alpha_W$ HPs, providing an opportunity to re-parametrize with a single HP. For instance, the scale of self-attention softmax activations is proportional to the product of $\alpha_W$ multipliers: $\mathrm{std}(x_{\mathrm{attn}}) \propto \alpha_{W_\mathrm{Q}} \alpha_{W_\mathrm{K}}$. In this instance we use a single $\alpha$ parameter (termed $\alpha_{\mathrm{attn\text{-}softmax}}$) and associate it with the attention operation, rather than the weights.

Table 3: Typical transformer HPs used under different schemes. *Basic* HPs in **bold** are most impactful and are commonly swept. *Extended* HPs in non-bold are not always swept, often set heuristically or dropped.

| SP | µP | u-µP |
|---|---|---|
| $\boldsymbol{\eta}$ | $\boldsymbol{\eta}$ | $\boldsymbol{\eta}$ |
| $\sigma$-scheme | $\boldsymbol{\sigma_{\mathbf{init}}}$ | |
| | $\boldsymbol{\alpha_{\mathbf{emb}}\vert\eta_{\mathbf{emb}}}$ | $\alpha_{\mathrm{ffn\text{-}act}}$ |
| | $\alpha_{\mathrm{attn}}$ | $\alpha_{\mathrm{attn\text{-}softmax}}$ |
| | $\alpha_{\mathrm{out}}$ | $\alpha_{\mathrm{res}}$ |
| | base-width | $\alpha_{\mathrm{res\text{-}attn\text{-}ratio}}$ |
| | base-depth | $\alpha_{\mathrm{loss\text{-}softmax}}$ |

We apply the same principle to all operations, unless they are unary and $k$-homogeneous for $k \geq 0$, in which case they propagate scale and don't require an HP (see Appendix G.1). This results in the set of HPs shown, with their placement in the model given in Table 7. We note that this procedure also works in other architectures than transformers, and naturally will produce other kinds of multipliers.

Thirdly, we use a single global $\eta$ and group $\alpha$ HPs across layers. This breaks our expressivity criterion, but we argue represents the best trade-off between expressivity and cardinality. We show in Appendix A.3 that having tuned a global $\eta$ HP and our extended $\alpha$ HPs, the further benefits of tuning per-tensor $\hat{\eta}_W$ HPs (which modify the global $\eta$) is minimal.

**Interdependency**. The second stage above, moving $\alpha$ HPs from weights into operations, not only reduces the number of HPs, but also minimizes the interdependence between those that remain. We find that u-µP's optimal HP values depend less on each other than under µP (see Section 4.1).

**Interpretability**. The combination of unit scale and reduced dependencies between HPs means that each $\alpha$ can be interpreted as determining some fundamental property of the model at initialization. For example, the $\alpha_{\mathrm{loss\text{-}softmax}}$ HP defines the (inverse of) the softmax's *temperature* for a unit-scaled input. We also introduce a new scaling scheme (defined in Appendix G.2.2) for residual connections, designed to give HPs independence and a clear interpretation: $\alpha_{\mathrm{res}}$ defines the contribution of the residual connections to the output scale, and $\alpha_{\mathrm{res\text{-}attn\text{-}ratio}}$ defines the relative contribution of attention versus FFN branches. Finally, we choose not to include base shape HPs in u-µP. They do not add to expressivity, lack a clear interpretation (besides alignment to a base model at a particular shape), break the interpretation of $\alpha$ HPs given above, and complicate implementation.

### 3.4 A NEW EMBEDDING LR RULE

Although theoretical transfer properties have been proved for µP, not all its HPs have had µTransfer shown empirically. We do so for the *extended* µP transformer HPs in Figure 15, where we observe

poor transfer across width for the embedding LR multiplier $\hat{\eta}_{\text{emb}}$. This suggests that the corresponding LR scaling rule for $c_{\text{emb}}$ is mis-specified.

We show in Figure 2 that changing it from the µP rule of $c_{\text{emb}} = 1$ to $1/\sqrt{\text{fan-out}}$ corrects this failure of HP transfer. As a result, we see improved loss under u-µP for larger model-sizes relative to µP. Our adoption of this change is a key factor in the improved performance of u-µP over µP in Figure 1. We offer no theoretical justification for this change, which we leave to future work.

### 3.5 HYPERPARAMETER SEARCH

As shown in section Section 2.1, the standard approach to HP search for µTransfer is via a random sweep over all HPs simultaneously. Sweeping individual HPs separately is challenging due to the dependencies between them. In contrast, u-µP's HPs are designed to admit such a strategy due to our interdependence criterion. Because of this, we propose a simpler sweeping strategy for u-µP which we term *independent search* (outlined in detail in Appendix A.5). Independent search involves a sweep of the LR, followed by a set of one-dimensional sweeps of the other HPs (which can be run in parallel). The best results from the individual sweeps are combined to form the final set of HP values.

## 4 EXPERIMENTS

Our experiments use the Llama architecture (Touvron et al., 2023a) trained on WikiText-103 (Merity et al., 2017) (except large-scale runs in Section 4.4). We apply best-practice LLM training techniques from the literature (see Table 5). In accordance with our analysis in Appendix C.1, we remove parameters from norm layers, use independent AdamW, and avoid training on too many epochs.

### 4.1 QUANTIFYING HYPERPARAMETER INTERDEPENDENCE

Our principled approach to HPs (see Section 3.3) contains the requirement that their optimal values should depend minimally on the value of other HPs. We now investigate this empirically, conducting a 2D sweep over every pair of HPs for µP and u-µP (see Figures 11 and 12 for pairwise results).

To derive an empirical measure of HP dependency, we introduce the notion of *transfer error* (see Algorithm 1). We take the best value of the transfer HP for each non-optimal value of the fixed HP, and use it with the optimal value of the fixed HP. The transfer error is the difference between the losses obtained and the minimum loss. Figure 3 shows this measure for each pair of HPs, reflecting the improvement in HP dependency as a result of our scheme. This gives u-µP a reduced risk of small transfer errors leading to large degradations, and the potential to sweep HPs in a more separable way.

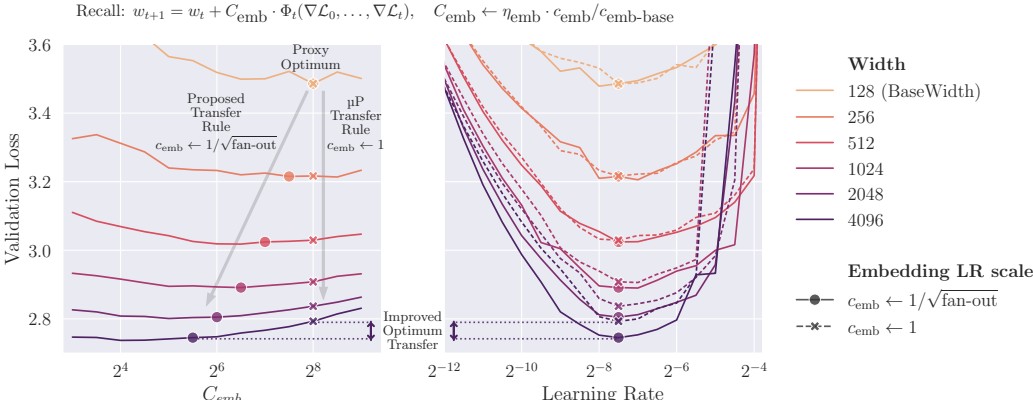

Figure 2: (Left) holding the embedding LR multiplier ($C_{\text{emb}}$) constant, vs. scaling with $\sqrt{1/\text{width}}$, both with a fixed global LR. This suggests the µP embedding LR rule ($c_{\text{emb}}$) should follow the latter scaling. (Right) we test this by sweeping the global LR under the two scaling rules. The new rule leads to lower loss on large models. (Dot/cross markers represent the same runs across both graphs).

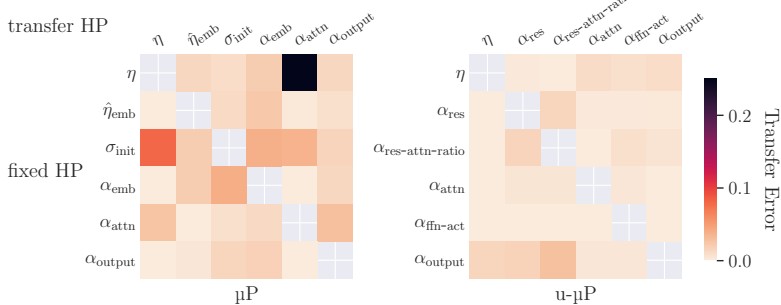

Figure 3: A visualization of the dependencies between pairs of HPs under each scheme. Transfer error

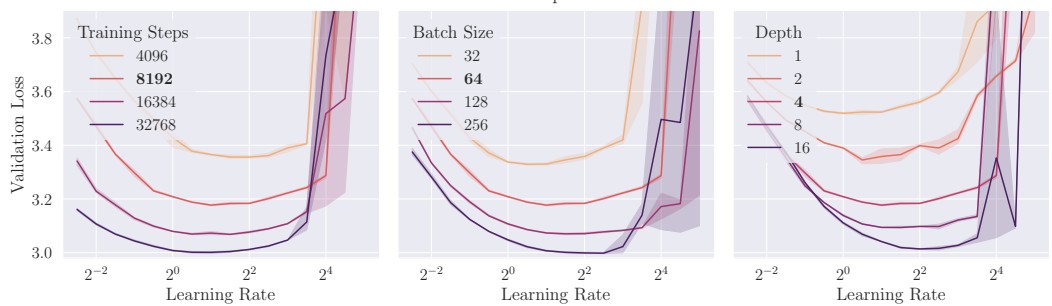

Figure 4: Learning rate transfer for u-μP over training steps, batch size and depth. See Figure 1 (b) for transfer over width and Figure 13 for regular μP transfer results. The **default** shape parameter for other panels is shown in bold. The shaded area shows the $95\%$ confidence interval for the mean.

## 4.2 HYPERPARAMETER SEARCH

Our new HP scheme, designed for improved separability, enables proxy model HPs to be swept more efficiently. This is shown in Figure 1 (a). We conduct a standard random search for μP and u-μP, along with the independent search outlined in Section 3.5 (and Appendix A.5).

Independent search begins with a simple LR sweep. This alone is sufficient for u-μP to reach near-optimal loss (using only 9 runs). During this phase other HPs are fixed at 1, which for u-μP means that the inputs to operations are generally unit-scaled. Consequently, we conclude that unit scale at initialization is close to ideal scaling for effective learning here. In contrast μP still requires non-LR HPs to be swept to attain a reasonable loss. The final 'combined mults' phase causes the loss to spike for μP. This is due to the HP dependencies shown in Figure 3, which mean HPs cannot be swept independently and used together, necessitating random search which can require hundreds of runs.

## 4.3 HYPERPARAMETER TRANSFER

We train many models and plot transfer of LR across width (Figure 1 (b)), steps, batch size and depth (Figure 4), and transfer of other HPs across width (Figure 15). Note that u-μP (building on μP) is designed to give transfer over width[3]; the other axes we report for practical purposes. We find that:

1. The optimal LR is constant across width under u-μP. There is a small drift for training steps and batch size, and a larger one with depth. Hence we recommend proxy models which primarily differ in width, moderately in steps and batch size, and least in depth.

2. Whereas μP sees diminishing returns for larger widths, u-μP continues to benefit from width. We attribute this primarily to our improved embedding LR rule.

3. Non-LR HPs also have constant optima across width under u-μP. This is not true for μP, where $\hat{\eta}_{\mathrm{emb}}$ has poor transfer (see Section 3.4), along with $\sigma_{\mathrm{init}}$ (see Appendix C.2).

4. The optimal values found for non-LR HPs are all close to 1. In practice this means that dropping these HPs entirely is potentially viable for similar models and training setups.

---

[3] As we use depth-μP this could be said about depth as well, but as Yang et al. (2023b) show that transformers don't attain depth-transfer under depth-μP we do not expect strong transfer across depth.

Table 4: 0-shot benchmark results at 7B scale.

| Scheme | Format | MMLU | HellaSwag | OpenBook QA | PIQA | TriviaQA | WinoGr |
|--------|--------|------|-----------|-------------|------|----------|--------|
| SP | BF16 | 29.6 | 52.4 | 27.8 | 76.5 | 22.2 | 63.3 |
| u-μP | BF16 | 29.0 | **53.4** | **31.6** | 77.1 | **23.4** | 63.7 |
| u-μP | FP8 | **31.2** | **53.4** | 29.6 | **77.6** | 21.3 | **65.7** |

## 4.4 FP8 TRAINING

In this section we justify the simple mixed-precision scheme described in Section 3.2 and demonstrate that it can be used to train u-μP models out-of-the-box.

**Proof-of-concept**. Figure 5 shows the RMS of all linear layer inputs for a moderately sized transformer. RMS captures the larger of the mean and scale of a distribution, and as such is a good test of whether a tensor is likely to suffer over/underflow in low-precision. We observe that u-μP tensors largely have RMS starting close to $1$ and remaining so at the end of training, supporting our scheme.

As an initial proof-of-concept we train a u-μP model using our FP8 scheme over 8k steps, using HPs from a proxy model, as shown in Figure 1 (c). We see only a small degradation versus FP32, and at this scale critical tensors can still be cast to FP8 using `E5M2`, while gradients can even use `E4M3`.

**Larger scale**. Next we consider a more realistic training scenario. Using the same architecture, and following the steps set out in our u-μP user-guide (Appendix D), we train our target models on 300B tokens of the SlimPajama dataset (Shen et al., 2023) (see Appendix A.8 for training details).

We begin with an independent search (Section 3.5) over our u-μP proxy model's HPs. Here we make the following observations:

1. When using a relatively small proxy model (8 layers and 512 width), the HP-loss landscape is rather noisy. By doubling the width we can discern optimal HP values more clearly.

2. The most important HPs are $\eta$ and $\alpha_{\text{res-attn-ratio}}$. All others can be left at the default of $1$.

3. The optimal values of these HPs are $\eta = 2^{3.5}$ and $\alpha_{\text{res-attn-ratio}} = 2^{-2.0}$ and thus differ non-trivially from the observed HPs in our smaller-scale experiments.

We then train u-μP models of approximately 1B, 3B and 7B parameters, using our FP8 mixed-precision scheme (see Section 3.2). We also train two baselines at each size: the first is a BF16 version of our u-μP models, and the second is a set of SP models using the weight init scheme from the Pythia model family (Biderman et al., 2023) and the LR scheme from Llama 3 (Dubey et al., 2024), scaling inversely with width and using a LR of 3e-4 at 7B scale. For the SP baseline, we use a non-independent weight decay of $0.1$ and parametric RMS norm as in Llama (Dubey et al., 2024), which is standard practice. The loss curves are shown in Figure 6. All FP8 runs converge and show no significant loss degradation. In comparison to SP, the u-μP models have a qualitatively different training curve with a higher loss for most of training that catches up in latter stages, hinting at a fundamentally different optimization trajectory. In terms of downstream performance, both of the u-μP 7B models are competitive with SP. In particular, the scores of the FP8 model are mostly on par with the BF16 models (see Table 4).

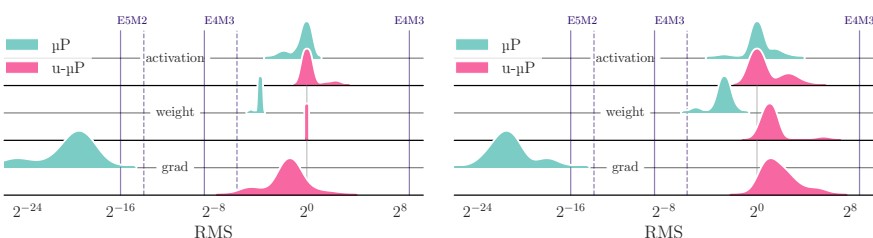

Figure 5: Per-tensor RMS $= \sqrt{\sigma^2 + \mu^2}$ across u-μP and μP models at initialization (left) and after training (right). Dashed and solid red lines show each format's min. normal and subnormal values.

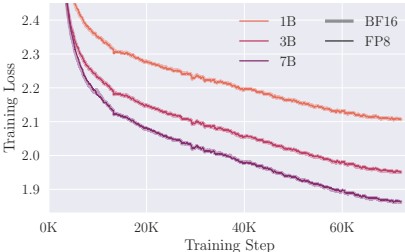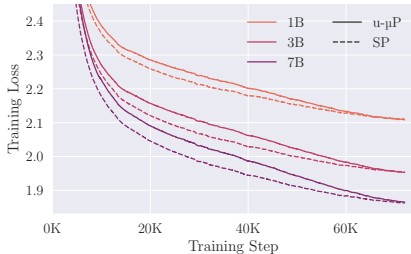

Figure 6: Large-scale training runs. (Left) u-μP BF16 vs u-μP FP8. (Right) u-μP BF16 vs SP BF16.

## 5 RELATED WORK

**Low-precision training**   Techniques to facilitate FP8 training include those covered in Appendix J and more (Wang et al., 2018; Mellempudi et al., 2019; Perez et al., 2023). These largely concern the quantizing of activations, weights and gradients, though Peng et al. (2023) also explore FP8 optimizer states and cross-device comms, which we consider interesting avenues of exploration. Quantization is made harder by the emergence of large outlier values when training at scale, with several techniques proposed to mitigate this (Bondarenko et al., 2023; Sun et al., 2024; He et al., 2024). Combining these with u-μP is a natural extension of our work, facilitating simpler or lower-bit quantization.

**Parametrizations**   The neural tangent kernel (NTK) (Jacot et al., 2018) and mean field parametrization (MFP) (Mei et al., 2018; Bordelon & Pehlevan, 2023) are alternatives to μP. In terms of width exponents, u-μP is identical to MFP except for the embedding learning rate scaling, as shown in Table 1 of Everett et al. (2024). The equivalence classes given by Everett et al. (2024) also mirror our notion of abc-symmetry. They additionally show that under MFP gradients become sufficiently small to require re-scaling the Adam epsilon term. u-μP naturally solves this problem in a similar way, via the up-scaling of our gradients to unit scale.

**Compatible parametrization frameworks**   Apart from u-μP, several other recent efforts have also introduced frameworks for training under particular parametrizations, which have the potential to be compatible with u-μP. Large et al. (2024) introduces the *modular norm* over the weight-space with the aim to ensure stable updates that provide LR transfer, like μP. Everett et al. (2024) explore the notion of *alignment* between parameters and data, showing that other parametrizations with per-layer LRs can outperform standard μP. These parametrizations could admit unit scaling under abc-symmetry, though this is outside the scope of this work.

**Signal propagation**   Unit Scaling and μP are mainly concerned about statistical properties of single tensors. Another line of work studies how the covariance between two distinct inputs propagates through the model. For MLPs this was analyzed in Poole et al. (2016); Schoenholz et al. (2017), revealing phase transitions in deep networks related to expressivity and trainability. For transformers, more recent work (Noci et al., 2022) studies the collapse of token representations. Coming from this angle, they derive the same $1/\sqrt{\mathrm{depth}}$ residual multiplier scaling that is used in u-μP.

## 6 CONCLUSIONS

We introduce u-μP, a modified and improved version of μP that satisfies Unit Scaling. Through careful analysis guided by first principles we identify an interpretable set of HPs that has minimal interdependencies and facilitates an efficient independent sweeping strategy. We show that the stability properties of μP combined with Unit Scaling enable a simple and robust FP8 mixed precision scheme that works in a realistic large scale training scenario.

**Limitations and future work**. Some choices like the modified embedding LR rule are only justified by empirical observations, and currently lack a theoretical explanation. Additionally, neither μP nor Unit Scaling give guarantees for network quantities to be well-behaved over the course of training. In particular we would like to understand the issue (or feature) of scale growth in the critical layers better and look into possible mitigations. We also believe that low-precision training techniques can be pushed further, with u-μP offering an ideal starting point for future optimizations.

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

## A  ADDITIONAL EXPERIMENTAL DETAILS

### A.1  EXPERIMENTAL SETUP

Our experimental analysis of u-µP was conducted by adapting the codebase used for Tensor Programs V, allowing us to compare µP and u-µP in the same setting. We change various experimental settings from the µP paper to make our experiments better reflect standard training procedures, particularly the dataset which we switch from WikiText-2 to the larger WikiText-103 (Merity et al., 2017). Where not specified otherwise, the default setting used in our experiments are given in Table 5. These also represent the settings of our proxy model.

| | |
|---|---|
| Dataset | WikiText-103 (Merity et al., 2017) |
| Sequence length | 256 |
| Vocab size | 32000 |
| Training set tokens | 138M |
| Architecture | Llama (Touvron et al., 2023a)  (Transformer, PreNorm, RMSNorm, SwiGLU, RoPE, "untied" embeddings), non-trainable RMSNorm parameters. |
| Width | 256   (scaled up to 4096) |
| Depth | 4 |
| Number of heads | 4   (scaled up to 64) |
| Head dimension | 64 |
| Total parameters | $19.5M$   (scaled up to 1.07B) |
| Batch size | 64 |
| Training steps | 8192 (0.97 epochs) |
| LR schedule | Cosine to $10\%$, 2000 steps warm-up |
| Optimizer | AdamW $(\beta_1, \beta_2, \epsilon) = (0.9, 0.999, 10^{-8})$ |
| Weight decay | $2^{-13}$, independent (Loshchilov & Hutter, 2019) |
| Dropout | 0.0 |
| µP HP search range | $\eta \in [2^{-10}, 2^{-6}]$ |
| | $\hat{\eta}_{\text{emb}} \in [2^0, 2^8]$ |
| | $\sigma_{\text{init}}, \alpha_{\text{emb}}, \alpha_{\text{attn}}, \alpha_{\text{output}} \in [2^{-2}, 2^2]$ |
| u-µP HP search range | $\eta \in [2^{-1}, 2^3]$ |
| | $\alpha_{\text{attn}} \in [2^{-2}, 2^2]$ |
| | $\alpha_{\text{residual}}, \alpha_{\text{residual-attn-ratio}}, \alpha_{\text{ffn-act}}, \alpha_{\text{output}} \in [2^{-3}, 2^3]$ |
| µP HP defaults | $\sigma_{\text{init}} = \alpha_{\text{emb}} = \alpha_{\text{attn}} = \alpha_{\text{output}} = \hat{\eta}_{\text{emb}} = 1$ |
| u-µP HP defaults | $\alpha_{\text{residual}} = \alpha_{\text{residual-attn-ratio}} = \alpha_{\text{ffn-act}} = \alpha_{\text{output}} = \alpha_{\text{attn}} = 1$ |

Table 5: Default hyperparameters and training settings.

## A.2 VALIDATING OUR EXPERIMENTAL SETUP

In this section we run a series of ablations to validate decisions relating to our experimental setup given above. In particular, we examine the effect of using repeated data, the effect of using a shorter warmup duration, and the effect of different final learning rates at the end of decay.

### A.2.1 REPEATED DATA

As outlined in Table 5, our standard training setup uses 0.97 epochs of the WikiText-103 dataset (50x larger than the WikiText-2 dataset used in Tensor Programs V). However on our batch size and training steps scaling experiments in Figure 4 we train on up to $4\times$ the amount of data than in our standard setup, and hence use up to $4$ epochs.

Though this is still a small level of repeated data, this moves our training slightly into the over-fitting regime. Based on this change, we here investigate the hypothesis that this regime has better or worse transfer of the optimal LR than the non-overfitting regime, and hence our results could be misleading. To do so, we repeated these experiments with the same number of tokens, but using the much larger SlimPajama dataset (Shen et al., 2023) where we use $< 1$ epoch.

The results for this experiment are seen in Figure 7. The shape of curves is very similar across the two datasets, for both batch size and training steps (albeit with a higher loss, due to the more varied nature of SlimPajama). From this we conclude that the effect of repeated data from our use of WikiText-103 is not significant.

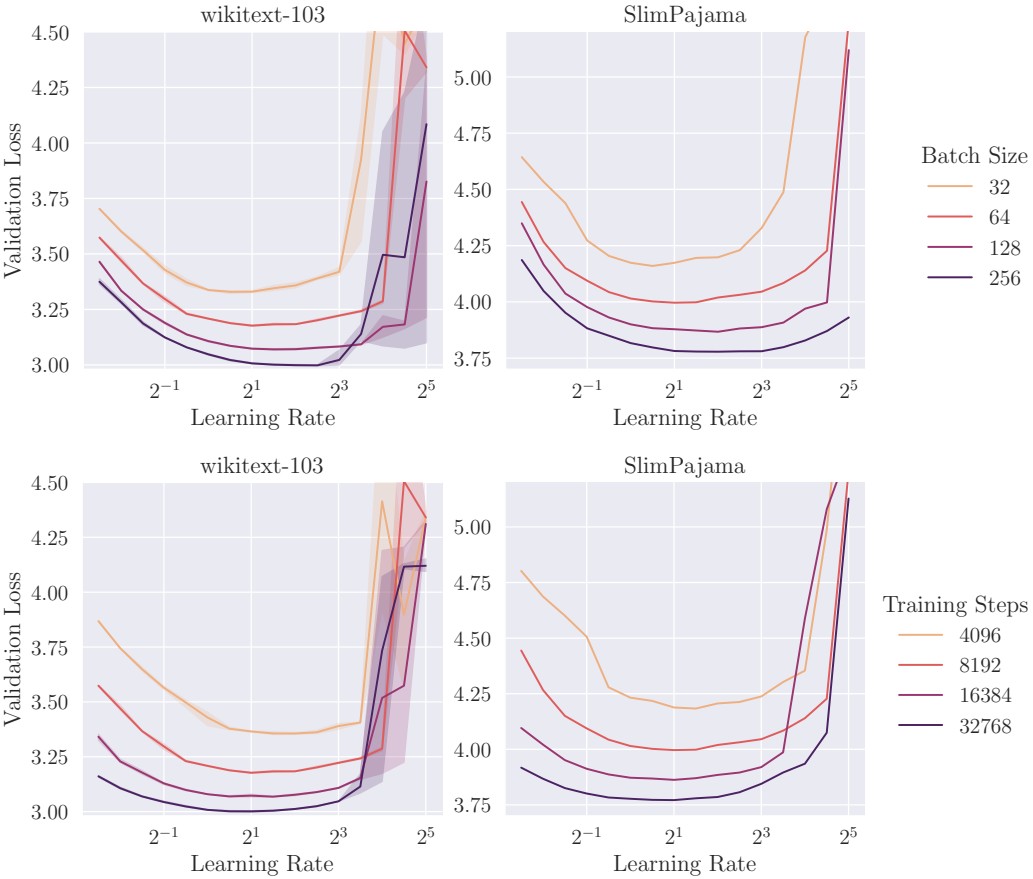

Figure 7: A repeat of the batch size and training steps experiments in Figure 4, but using the larger SlimPajama dataset where no data is repeated. In both settings our validation loss basins take the same shape, indicating that our analysis using the WikiText-103 dataset holds.

### A.2.2 Warmup duration

For our experimental setup (Table 5) we use a longer duration of warmup than in our large-scale setup (Table 6). We do so out of caution, as we use fewer tokens-per-batch for the smaller-scale experiments and so may require longer warmup. However, doing so also creates the risk of spending too large a proportion of training doing warmup, which could affect transfer.

To investigate this effect, we run two experiments. Firstly, we re-run the experiment for LR transfer over training steps, shown in Figure 4, on a quarter of the warmup steps. This is shown in Figure 8 (left). The main effect appears to be higher loss for larger learning rates, but the optima are largely unchanged. The only exception is the 4096-step run, where the optimum shifts left and the loss improves slightly. This appears to now align the optimum better with the other training durations, but leads to narrower basins as a result, suggesting a trade-off for this particular experiment.

However, all our other experimental runs use the 8192-step configuration, which has a consistent optimum regardless of warmup duration here. To investigate the effect of reduced warmup on width transfer at this particular step-count, we re-run our experiment in Figure 1 (b) under the shorter warmup duration, shown in Figure 8 (right). The only significant impact of this change is to narrow the basins, inducing no significant change in the optimal LR. As such, we conclude that using 2000 steps of warmup in our experimental setup is a reasonable choice, and both give the same width transfer.

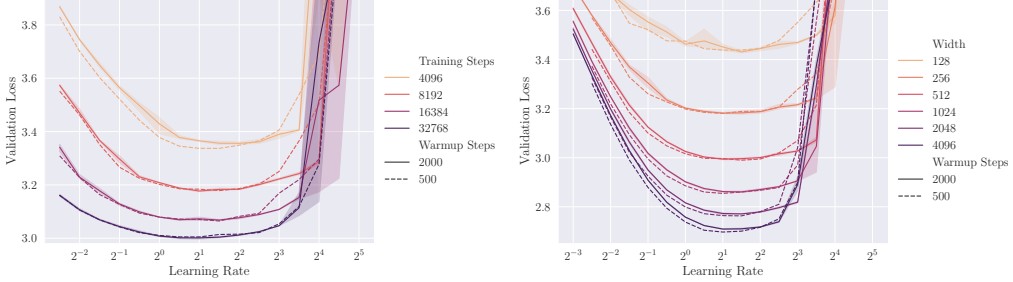

Figure 8: (Left) Learning rate transfer across training steps under different numbers of warmup steps. (Right) Learning rate transfer across width under different numbers of warmup steps. In this setting (training steps = 8192) the optimal LR is consistent, meaning either warmup regime can be used, though the longer gives wider basins.

### A.2.3 Learning rate decay target

In all our experiments we use a cosine decay of our learning rate down to 10% of the maximum. This follows the standard approach taken by most LLM training projects (Touvron et al., 2023b; Biderman et al., 2023; Groeneveld et al., 2024; Almazrouei et al., 2023; Yang et al., 2024). However, recent research has indicated that this may not be the optimal decay target, with implications for LR transfer. Hägele et al. (2024) show that the choice of target percentage can alter the shape of transfer curves and potentially shift the optimum value (Figure 21, right). They also suggest that using a fixed target value may work better than a percentage (Figure 22, right), which could be swept separately. Anonymous (2024) separately suggest that linear decay to zero is the most effective scheme.

Though using the optimal decay scheme is not necessarily essential to the validity of our method, any implications of different schemes on transfer properties should be investigated. To do so, we run two experiments. The first sweeps the learning rate for our standard model at various percentages and fixed values of cosine decay target, including zero, in Figure 9 (left). Lower decay targets perform better here, including zero, suggesting that this simple rule may be ideal.

We then re-run our width transfer experiment from Figure 1 (b) but with our LR decaying to 0, and plot the result in Figure 9 (right). This leads to slightly better results for large learning rates, though for large models this difference diminishes. Fortunately the effect this decay target has on the shape of curves (and hence optimal LR transfer) is minimal, indicating that our conclusions are not effected significantly by the choice of decay target.

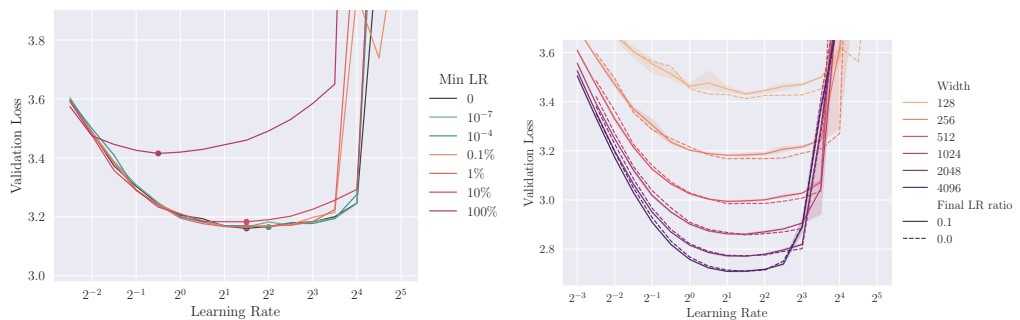

Figure 9: (Left) A learning rate sweep over LR targets of different types (percentage, fixed and zero) on our standard model. (Right) Using the zero and 10% learning rate targets, LR transfer over width.

A.3 PER-TENSOR LEARNING RATES

In Section 3.3 we relax the requirement for each weight tensor in the u-µP model to have an associated tuneable learning-rate multiplier on top of the global learning rate. Whilst this does reduce the theoretical expressivity of the u-µP scheme, Figure 10 shows that using a single globally optimized learning rate is already at or close to the optimal choice for all weight tensors, and therefore it is reasonable to drop these multipliers in favor of reducing the number of HPs. However, a practitioner attempting to absolutely maximize the task performance of their model could experiment with tuning a few key per-tensor LRs, in particular the embedding table.

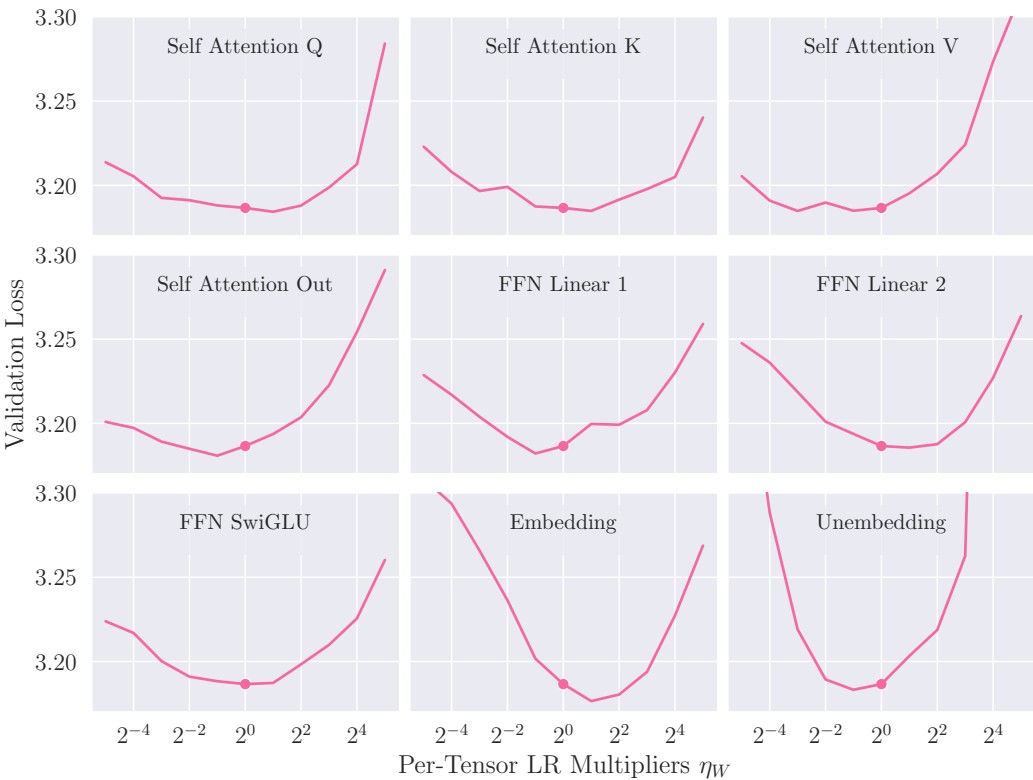

Figure 10: Independently varying per-tensor learning rate multipliers $\eta_W$, using the u-µP model of width 256 from Figure 1 with optimized global learning rate $2^{1.5}$ as the starting point. Where applicable, the same multiplier is used for tensors of the same name across transformer layers. Each subplot fixes all but one multiplier at 1, therefore the midpoint of each subplot is precisely the u-µP$_{256}$ model from Figure 1.

A.4 HYPERPARAMETER INDEPENDENCE

In Section 4.1 we explore the question of HP independence under µP and u-µP. The following plots in Figures 11 and 12 show the result of a 2D sweep over every pair of HPs under each scheme. All other HPs are held at 1 when not swept, except the $\eta$ which is held at $2^{-7.5}$ for µP and $2^{1.5}$ for u-µP, and $\hat{\eta}_{\text{emb}}$ which is held at $2^4$ for µP.

These results show visual dependence between µP hyperparameters as a diagonal structure in the grids, such as $(\hat{\eta}_{\text{emb}}, \sigma_{\text{init}})$ and $(\eta, \alpha_{\text{attn}})$. We quantify this in the plot in Figure 3, where we use a measure of HP dependence termed transfer error. This is explained verbally in Section 4.1, and we provide an algorithmic description in Algorithm 1. We note that differences in transfer error between the two methods may also be influenced by the flatness of the optimum. The HP and loss values used for our transfer error calculations are those in Figures 11 and 12.

---

**Algorithm 1** Transfer Error

---

**Require:** A 'fixed' HP with candidate values $F = \{f_1, \cdots, f_n\}$, a 'transfer' HP with candidate values $T = \{t_1, \cdots, t_m\}$, a function that gives the final validation loss for the pair of HPs $L : F \times T \to \mathbb{R}$ (assuming all other HPs are fixed at default values).

$\quad$ err $\leftarrow 0$
$\quad$ $f^*, t^* \leftarrow \text{argmin}(L)$
$\quad$ **for** $f$ in $F$ **do**
$\quad\quad$ **if** $f \neq f^*$ **then**
$\quad\quad\quad$ $t \leftarrow \text{argmin}(L(f))$
$\quad\quad\quad$ err $+= L(f^*, t) - L(f^*, t^*)$
$\quad\quad$ **end if**
$\quad$ **end for**
$\quad$ **return** err$/(n-1)$

---

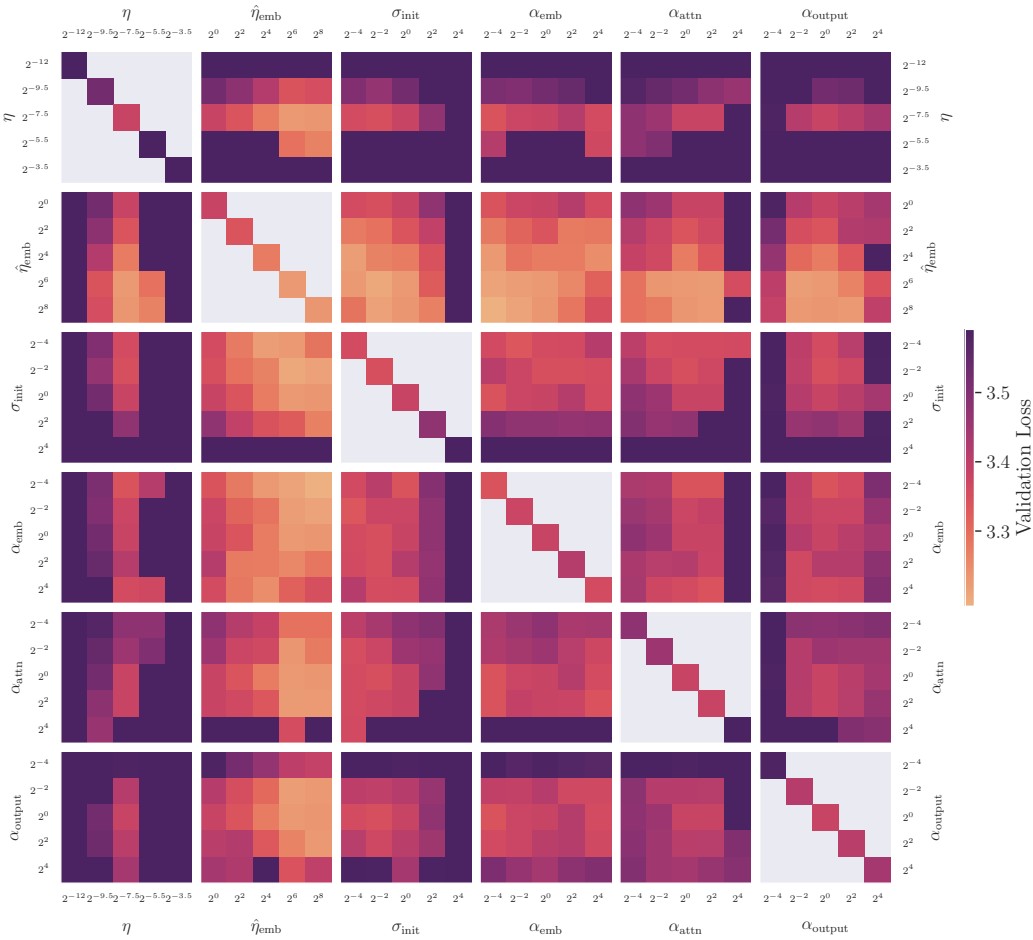

Figure 11: Hyperparameter coupling sweep for μP. Note strong coupling between optima, e.g. in the cases of $(\hat{\eta}_{\text{emb}}, \sigma_{\text{init}})$ and $(\eta, \alpha_{\text{attn}})$. See also: u-μP, Figure 12.

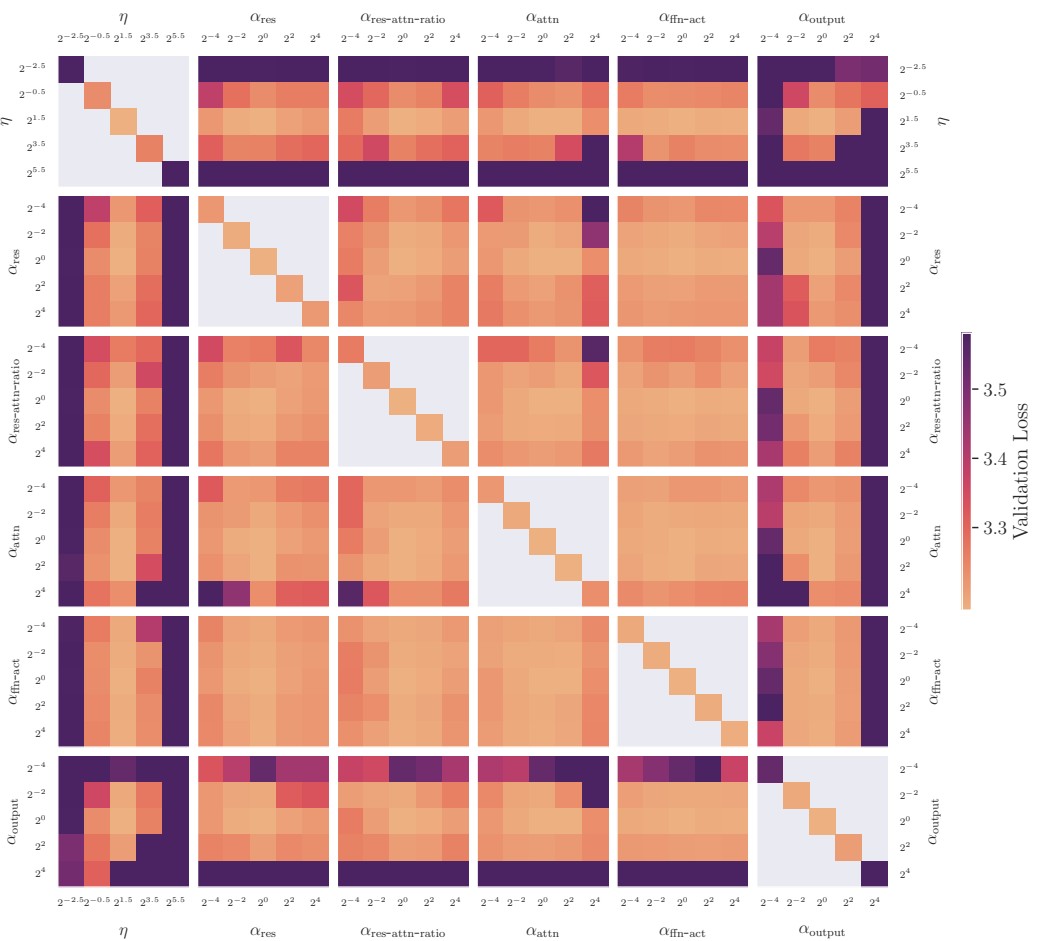

Figure 12: Hyperparameter coupling sweep for u-μP. Note less coupling than with μP, see Figure 11.

A.5    HYPERPARAMETER SEARCH

Here we outline the particular search processes used for our µP and u-µP HP sweeps in Figure 1 (a). The *random search* samples uniformly from a grid defined over all *extended* HPs (extended HP sets are defined in Table 3, with grid values defined in Table 5). We perform the random search over 339 runs, each of which is a full training of the width-256 proxy model. We then simulate the effect of shorter searches at various run-counts by taking a random sample of the results, resulting in the smooth curve over run-count shown.

The *independent search* consists of the following phases:

1. Perform a 1D line search for an optimal learning rate, with other hyperparameters set to their default values (9 runs).
2. For each hyperparameter in parallel, perform a 1D line search (330 runs).
3. Combine the best settings from step 2, and re-evaluate (6 runs).

The number of runs in the 1D line search is an order of magnitude higher than is required in practice. We do so to form a fair comparison with the random search, which benefits from this large number of runs. The number of runs for the 1D line search could be reduced further by using binary search, though this would require sequential runs and limit the extent of parallelism.

A.6    HYPERPARAMETER TRANSFER EXPERIMENTS

**Baseline µP transfer**    Figure 13 is a companion plot to Figure 4 in the body of the paper, showing the LR transfer of the baseline µP model over the same axes. u-µP shows marginally more stable HP transfer here relative to the baseline, and at a consistently lower loss.

**LR transfer over width**    The transfer experiments shown in Figure 1 (b) use the non-LR HPs found in Figure 1 (a) (indicated by the circled points), rather than using default HP values. For the u-µP sweep we take the HPs at the end of the LR portion of the independent search, as these are already close-to-optimal, and means only 9 runs were required in the sweep. In contrast, for µP it is necessary to use the results of the random search over a large number of runs.

**LR transfer over other axes**    For the training steps, batch size and depth transfer experiments in Figure 4, all HP values are fixed to 1 except LR which is swept. As with width transfer, u-µP outperforms µP here using these default HP values. Reducing training steps is done by fixing the number of warm-up steps (at 2000) and still cosine-decaying the learning rate to $10\%$; all that changes is the number of post-warm-up steps. We found this to be more effective than cutting-short the decay schedule. For both Figure 1 (b) and Figure 4 we sweep the LR over a logarithmically-spaced grid of step $2^{1/2}\times$, with 3 runs for each point.

Additionally, in Figure 14 we show learning rate transfer over sequence length for both µP and u-µP fixing either tokens per batch or sequences per batch. In both scenarios u-µP shows not only better absolute training performance, but also better transfer behaviour as sequence length increases. Since our default proxy sequence length is 256, using µP to transfer to sequence length 2048 would result in minimal improvements or even a degradation in validation loss, whereas the u-µP shows much greater and more consistent improvements.

**Other HP transfer over width**    For our non-LR HP transfer results in Figure 15, we note that good transfer under µP has not been demonstrated for all HPs used in the literature. This is particularly true for the $\hat{\eta}_{\mathrm{emb}}$ HP, which has poor transfer under µP. Our investigation here led to our identification of the need to adjust the embedding LR scaling rule outlined in Section 3.4. In many cases users have not swept this HP, but instead swept the corresponding parameter multiplier $\alpha_{\mathrm{emb}}$. How this HP interacts with the embedding LR scaling problem identified (and our proposed fix) remains to be explored, though we note in Figure 15 that it also appears to have poor transfer.

**Combined HP transfer**    Whilst Figure 15 demonstrates the transfer of individual hyperparameters over width, Figure 16 instead demonstrates the simultaneous transfer of all hyperparameters when co-optimized on the small-scale proxy model, as is done for µTransfer. The µP and u-µP points are

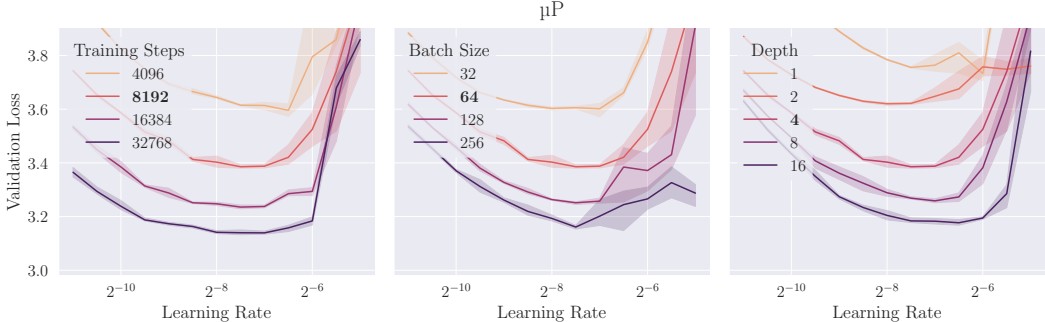

Figure 13: Learning rate transfer for μP over training steps, batch size and depth. This is a companion to the equivalent u-μP plot in Figure 4.

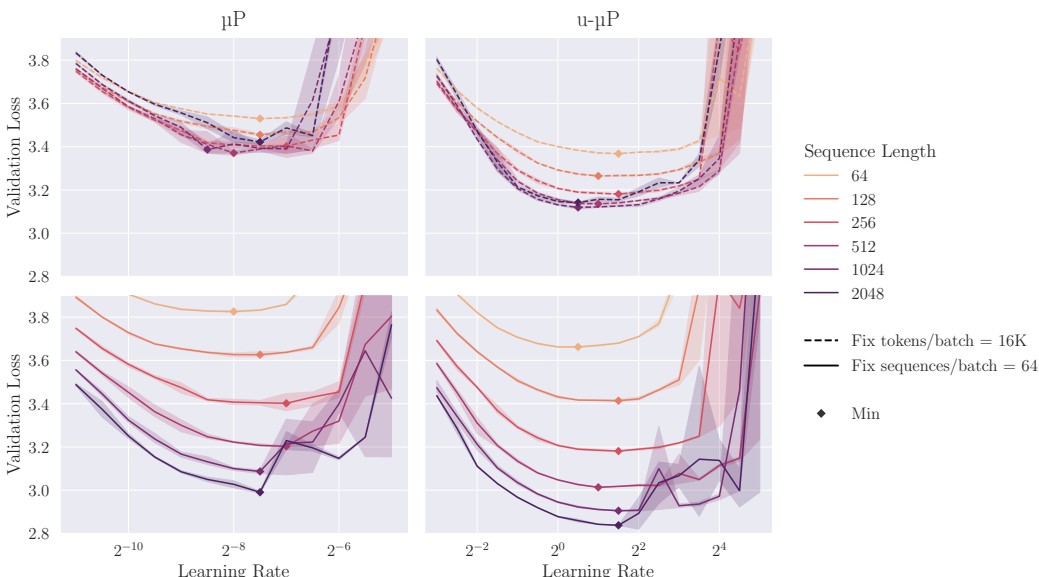

Figure 14: Transfer of learning rate over sequence length for μP (left) and u-μP (right). As sequence length varies, we can fix the number of tokens per batch by inversely varying the number of sequences per batch (top). Alternatively we can fix the sequences per batch and allow the number of tokens per batch to vary with sequence length (bottom). In the latter case, larger sequence lengths mean the model sees more tokens during training, though as per Table 5 this translates to >1 epoch on WikiText-103 when sequence length goes above 256.

taken from Figure 1, with hyperparameters swept on a model of width 256 using a full random HP search and a simple learning rate sweep for μP and u-μP respectively. The Standard Parametrization scheme, as shown in Table 3 requires choosing a learning rate and a weight-initialization scheme. We follow the initialization scheme of Pythia (Biderman et al., 2023), and transfer learning rate using a heuristic scaling factor of $^{\text{base-width}}/_{\text{width}}$, as is done in Dubey et al. (2024).

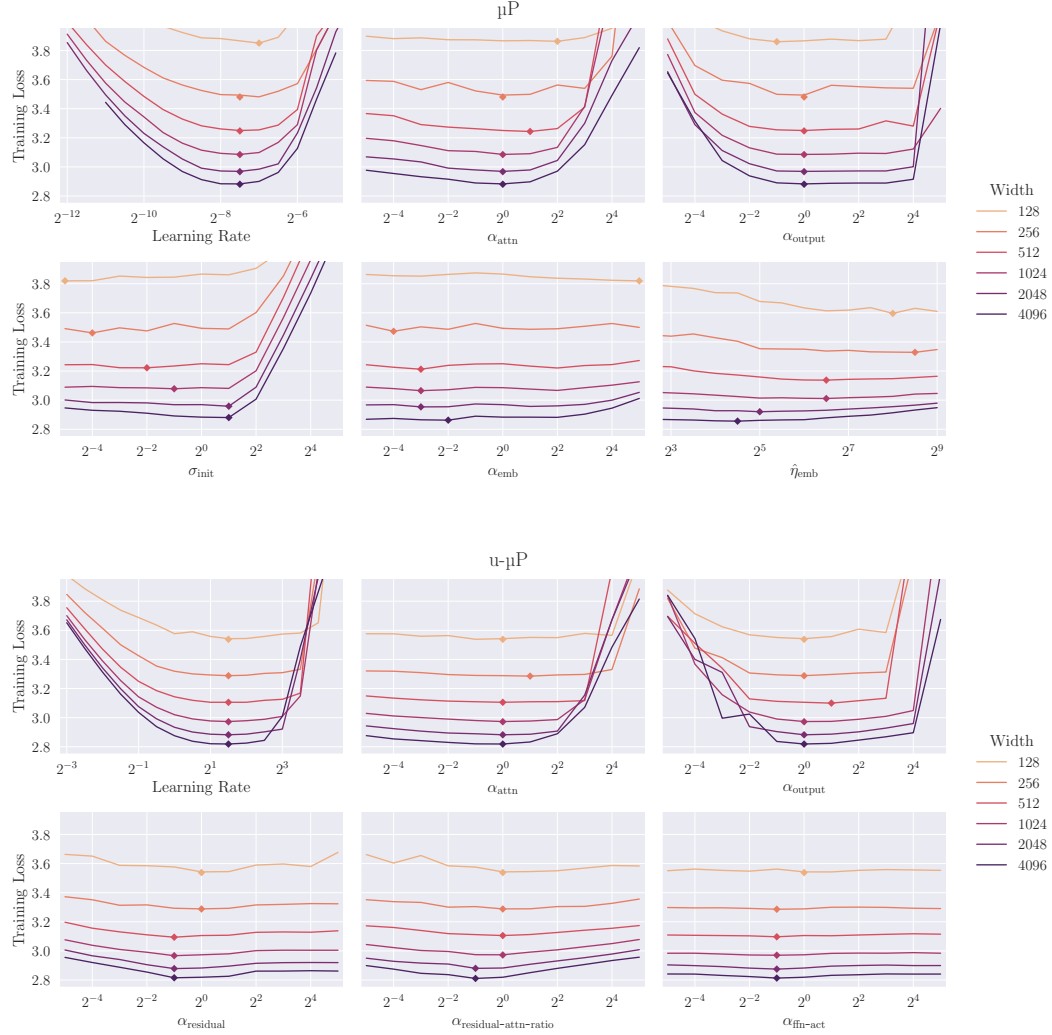

Figure 15: Transfer of model hyperparameters over width for μP (top) and u-μP (bottom). When one hyperparameter is being swept, all others are fixed at 1, with the exception of Learning Rate $\eta = \left(2^{1.5}, 2^{-7.5}\right)$ for (u-μP, μP).

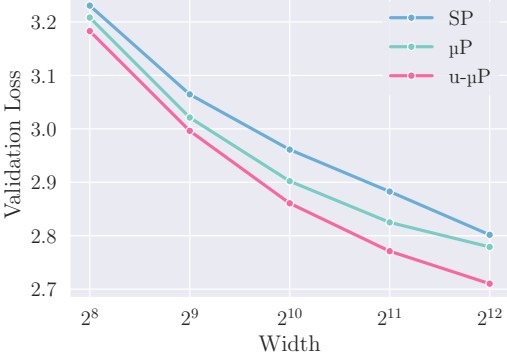

Figure 16: Transferring hyperparameters from width 256 up to 4096 using three different hyperparametrization schemes. μP and u-μP results are as seen in Figure 1, whilst Standard Parametrization follows the initialization approach of Pythia (Biderman et al., 2023).

### A.7 NUMERICAL PROPERTIES

Our analysis of the numerical properties of u-μP focuses on the RMS of tensors that we wish to cast to FP8: linear module input activations, weights and output gradients. From the RMS training statistics plots in Figure 5 and Figure 17 we note that

1. μP has gradients and weights with low RMS, at risk of FP8 underflow, whereas u-μP starts with $\mathrm{RMS} \approx 1$.

2. Many input activations do not grow RMS during training (due to a preceding non-trainable RMSNorm), however the attention out projection and FFN down projection have unconstrained input activations that grow considerably during training.

3. The decoder weight grows during training. Since it is preceded by a RMSNorm, the model may require scale growth in order to increase the scale of softmax inputs. Other weights grow slightly during training.

4. Gradients grow quickly but stabilize, except for attention out projection and FFN down projection, whose gradients shrink as the inputs grow.

We also evaluate how RMS growth is affected by model and training hyperparameters in the tensors that showed the highest end-training RMS, shown in Figure 18. This shows that the main parameter affecting scale growth is learning rate, with end-training RMS increasing to the right of the optimal LR basin, as training becomes unstable. End-training RMS is remarkably stable as width, depth, training steps and batch size are independently increased.

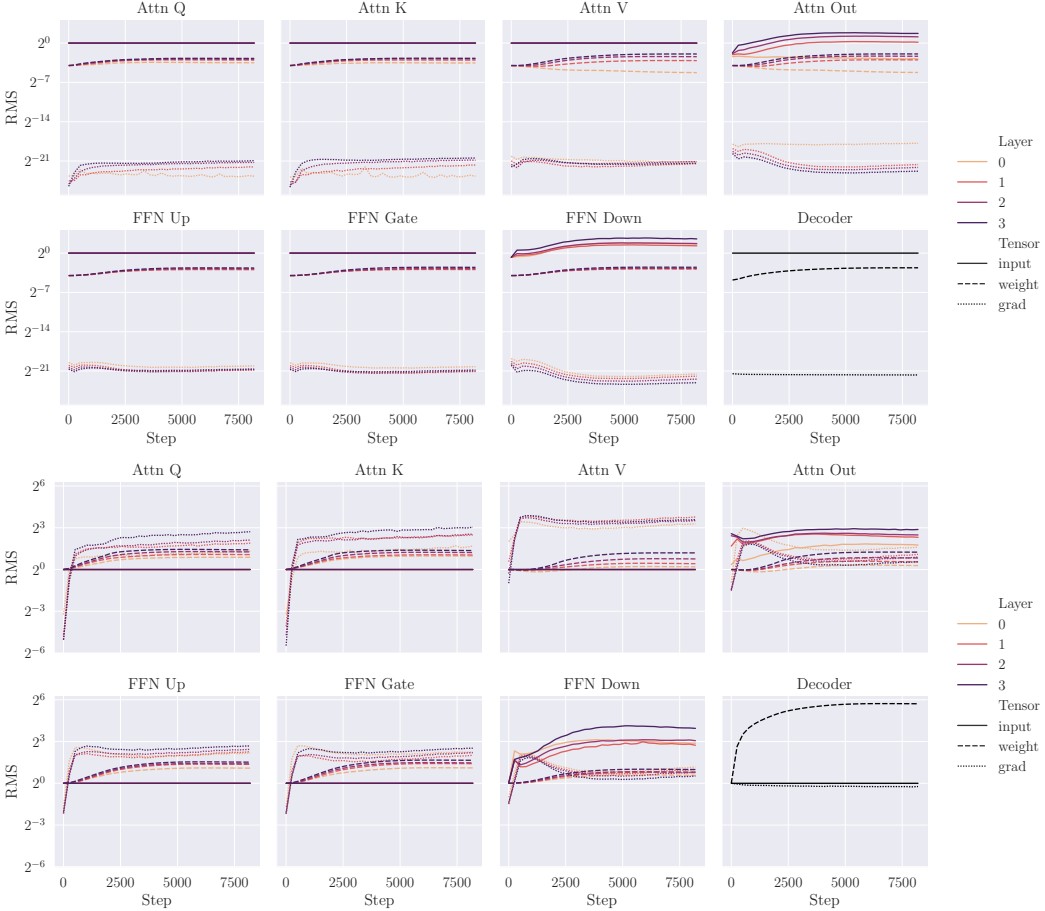

Figure 17: RMS during training, for all parametrized matmul inputs, for μP (top) and u-μP (bottom). Model width 256, default hyperparameters, $\eta = (2^1, 2^{-8})$ for (u-μP, μP).

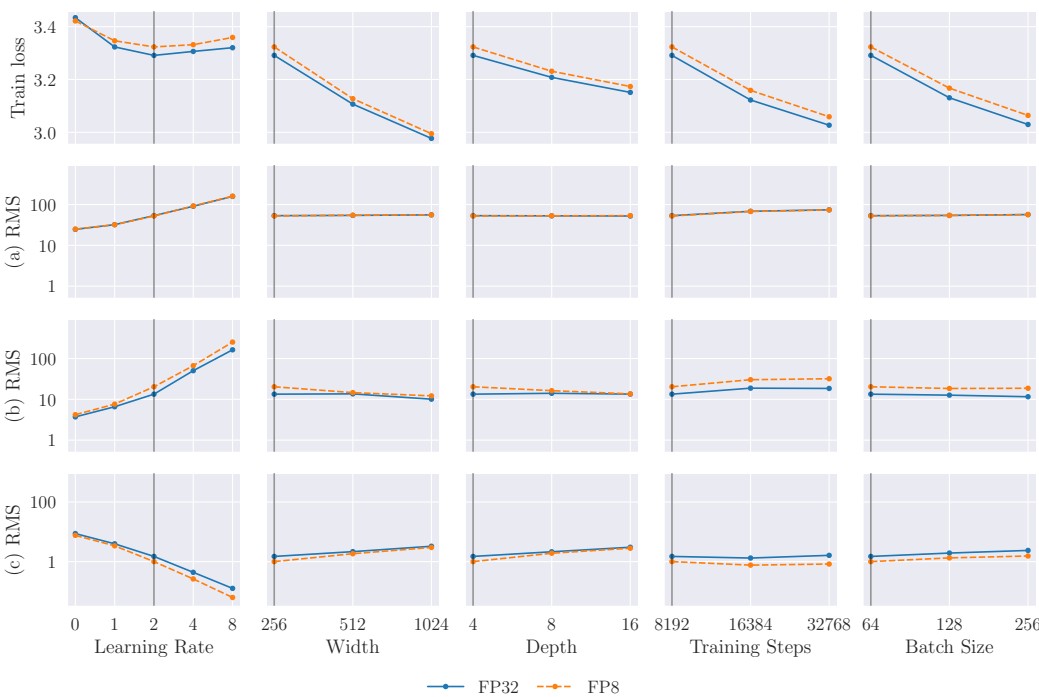

Figure 18: The effect of hyperparameters on FP8 training loss and on the end-training RMS of critical tensors: (a) decoder weight, (b) last-layer FFN down-projection input and (c) last-layer FFN down-projection output gradient. Only learning rate has a substantial effect on the end-training RMS. Vertical lines show the default setting of that hyperparameter, as used for all other plots.

## A.8 LARGE-SCALE TRAINING DETAILS

Our large-scale training settings are given in Table 6. These are largely the same as our standard experiments (Table 5), but with many more tokens used for training and scaling up to a larger model-size.

| | |
|---|---|
| Dataset | SlimPajama (Shen et al., 2023) |
| Sequence length | 4096 |
| Vocab size | 65536 |
| Training set tokens | 600B |
| Architecture | Llama (Touvron et al., 2023a) (Transformer, PreNorm, RMSNorm, SwiGLU, RoPE, "untied" embeddings), non-trainable RMSNorm parameters. |
| Width | $[2048, 3072, 4096]$ (1024 for proxy model) |
| Depth | $[16, 24, 32]$ (8 for proxy model) |
| Number of heads | $[16, 24, 32]$ (8 for proxy model) |
| Head dimension | 128 |
| Total parameters | $[1.07B, 3.12B, 6.98B]$ |
| Batch size | 1024 |
| Training steps | 72000 ($\sim$ 300B tokens; 20000 for proxy model) |
| LR schedule | Cosine to $10\%$, 500 steps warm-up |
| Optimizer | AdamW $(\beta_1, \beta_2, \epsilon) = (0.9, 0.95, 10^{-8})$ |
| Weight decay | $2^{-13}$, independent (Loshchilov & Hutter, 2019) |
| Dropout | 0.0 |

Table 6: Large-scale training settings.

We use mixed-precision during training with optimizer states in FP32 that are sharded via ZeRO stage 1 (Rajbhandari et al., 2020). We retain the model weights in BF16 and apply our FP8 scheme as described in Section 3.2 to the tensors participating in matmul operations throughout the transformer block. All other tensors remain either in BF16 (embedding, readout layer, norm, activation function) or FP32 (Flash Attention (Dao et al., 2022)).

Each model was trained on several Nvidia A100 (80GB) or H100 GPUs, with all FP8 experiments conducted on the H100 chips utilizing their native FP8 support. For the FP8 operations we use PyTorch's `torch._scaled_mm` function as a backbone.

# B  UNIT-SCALED OP DEFINITIONS

Table 7: Implementations of unit-scaled ops, building on Table A.2. from the Unit Scaling paper (Blake et al., 2023). These are considered part of u-μP and should be used in the place of standard operations.

| Op | Unit Scaling factors |
|---|---|
| $\mathrm{matmul}(x, w) = xw$ | $\alpha = \frac{1}{\sqrt{\text{fan-in}}}, \beta_x = \frac{1}{\sqrt{\text{fan-out}}}, \beta_w = \frac{1}{\sqrt{\text{batch-size}}}$ |
| $\mathrm{attention}(q, k, v) =$ $\mathrm{softmax}\left(\alpha_{\text{attn}} \, d_{\text{head}}^{-1} \left(qk^\top\right) \odot c_{\text{mask}}\right) v$ | $\alpha = \beta_q = \beta_k = \beta_v =$ $1/\log\_\mathrm{interpolate}\left(\frac{1}{1+\frac{4d_{\text{head}}}{\alpha_{\text{attn}}^2}}, 1, \sqrt{\frac{\log(s)}{s}}\right)$ |
| $\mathrm{gated\_silu}(x_{\text{in}}, x_{\text{gate}}) =$ $x_{\text{in}} \odot x_{\text{gate}} \odot \mathrm{sigmoid}(\alpha_{\text{ffn-act}} \, x_{\text{gate}})$ | $\alpha = \beta_{x_{\text{in}}} = \beta_{x_{\text{gate}}} =$ $1/\log\_\mathrm{interpolate}\left(\frac{1}{1+\frac{1}{\alpha_{\text{ffn-act}}^2}}, \frac{1}{\sqrt{2}}, \frac{1}{2}\right)$ |
| $\mathrm{residual\_add}(x_{\text{resid.}}, x_{\text{skip}}) =$ $a \, x_{\text{resid.}} + b \, x_{\text{skip}}$ | $a = \frac{\tau}{\sqrt{\tau^2+1}}, b = \frac{1}{\sqrt{\tau^2+1}}$ (see G.2.2 for full details, inc. values for $\tau$, which depends on $\alpha_{\text{res}}$ and $\alpha_{\text{res-attn-ratio}}$.) |
| $\mathrm{softmax\_xent}(x, t) =$ $\log\_\mathrm{softmax}(\alpha_{\text{loss-softmax}} \, \mathrm{x})_t$ | $\alpha = 1, \; \beta = s/\sqrt{s-1}$ |
| $\mathrm{RoPE}(x)$ | $\alpha = \beta = 1$ (i.e. no scaling) |
| $\mathrm{RMSNorm}(x)$ (non-trainable, see (Lingle, 2024)) | $\alpha = \beta = 1$ (i.e. no scaling) |

The Unit Scaling paper provides scaling factors for various ops, in order to make them unit-scaled. However, these ops do not cover every case required for the Llama architecture used in our experiments, nor do they cover our updated residual layer implementation. To address this, in this section we outline a series of new unit-scaled ops for each of our required architectural features, as well as existing unit-scaled ops, as given in Table 7.

The presentation here is derived from that of the Unit Scaling Compendium given in (Blake et al., 2023, Table A.2). This makes reference to the factors $\alpha, \beta_1, \ldots, \beta_k$. $\alpha$ is the output scaling factor in the forward pass, and $\beta_i$ are the scaling factors for the gradient of the op's inputs in the backward pass. For each op, a value or rule is provided for determining the required mult to ensure unit-scale. The correct value for these multipliers is derived by analyzing the scaling behavior of each op, given some reasonable distributional assumptions about the input and incoming gradient tensors (see Appendix E.2 for an example). Below we provide an in-depth overview of each new or modified unit-scaled op we introduce here.

**Unit-scaled dot-product attention**  The Unit Scaling paper considers the attention layer scaling in terms of its separate components: the various matmul operations and the internal softmax. Linear operations are scaled using the standard rule, and the softmax scaling is given a $\alpha = \beta = s$ factor.

From an implementation perspective, the self-attention layer is more typically broken down into weight-matmuls and a fused scaled-dot-product attention operation. This is the case we handle here, accounting for three complicating factors not considered in the Unit Scaling paper:

1. As we use a decoder-style transformer in our experiments, our softmax operation has a causal mask applied to its input.

2. We follow the μP guidance of using $1/d_{head}$ scaling in our self-attention layer, rather than the usual $1/\sqrt{d_{head}}$.

3. We place a $\alpha_{\text{attn}}$ multiplier immediately before the softmax, which is an HP that users may tune.

As a result our dot-product attention takes the form:

$$\text{attention}(q, k, v) = \text{softmax}\left(\alpha_{\text{attn-softmax}} \cdot d_{\text{head}}^{-1} \cdot (q \cdot k^\top) \odot c_{\text{mask}}\right) \cdot v$$

The addition of an HP before the softmax introduces an additional challenge for Unit Scaling, as our scaling multipliers will need to account for this value when preserving unit scale.

This operation is sufficiently complex that we found an empirical model of its scale to be more accurate than any mathematically-derived rule (future work may consider justifying our model mathematically). We find that the scale of dot-product attention is approximately

$$\sigma(\text{attention}(q, k, v)) = \text{log\_interpolate}\left(\frac{1}{1 + \frac{4d_{\text{head}}}{\alpha_{\text{attn}}^2}}, 1, \sqrt{\frac{\log(s)}{s}}\right)$$

where

$$\text{log\_interpolate}(\alpha, b_{\text{upper}}, b_{\text{lower}}) = e^{\alpha \log(b_{\text{upper}}) + (1-\alpha) \log(b_{\text{lower}})}.$$

The corresponding scaling rule is therefore to divide by this factor in both the forward and backward pass, as outlined in Table 7.

**SwiGLU FFN**   Llama uses a SwiGLU (Shazeer, 2020) layer for its FFN, which introduces two new operations for us to unit-scale: a SiLU (Yu & Su, 2019) (a.k.a. swish (Ramachandran et al., 2018)) operation and an element-wise multiplication. We take a similar approach to our dot-product attention, and consider unit-scaling the following fused operation:

$$\text{gated\_silu}(x_{\text{in}}, x_{\text{gate}}) = x_{\text{in}} \odot x_{\text{gate}} \odot \text{sigmoid}(\alpha_{\text{ffn-act}} \, x_{\text{gate}})$$

For the surrounding weight-matmuls we follow the standard Unit Scaling rules.

Again, we use an empirical model of the scale of this op, which is surprisingly similar to the dot-product attention model:

$$\sigma(\text{gated\_silu}(x_{\text{in}}, x_{\text{gate}})) = \text{log\_interpolate}\left(\frac{1}{1 + \frac{1}{\alpha_{\text{ffn-act}}^2}}, \frac{1}{\sqrt{2}}, \frac{1}{2}\right),$$

dividing through by this factor to get our scaling rule.

**Residual layers**   Our implementation of residual layers for u-μP is more complex than other operations, as adjustments are required to:

1. Make pre-norm residual networks support Unit Scaling (see Appendix F).

2. Introduce our new, principled residual HPs (see Appendix G).

Our residual layer scheme is presented in full in G.2.2. For readers interested in our justification for this, see the sections noted above.

We also follow the example of Unit Scaling and delay the application of our residual multiplier in the backward pass to the base of the branch (see (Blake et al., 2023), Figure 3c). This does not change the model, and enables unit-scale to be maintained on the residual branch regardless of the value of the multiplier.

**RoPE embeddings**   We also require a unit-scaled implementation of Rotary Position Embeddings (RoPE (Su et al., 2024)), which are applied just before the scaled dot-product attention operation. As RoPE essentially consists of pair-wise rotations of elements by different degrees, we observe no meaningful scale-change as a result of it's application, and hence leave it unchanged.

**RMSNorm**   Following (Lingle, 2024) we opt to use a non-trainable version of RMSNorm (Zhang & Sennrich, 2019), in order to facilitate better transfer. As a result, we also leave this operation unchanged. Were a trainable RMSNorm to be used, the recipe would follow closely that of the LayerNorm presented in the Unit Scaling paper's compendium.

**Scale constraints**   One final, minor deviation from the scheme outlined in the Unit Scaling paper is the way in which we apply scale constraints (see their Section 5.2). The essence of scale constraints is that for perfect unit scaling, sometimes the ideal scale for the forward pass differs from those in the backward pass. In some special cases (e.g. at the ends of the network) the use of different scales can be valid, but in the general case a single scale must be agreed upon. The solution in the Unit Scaling paper is to use the geometric mean of the forward and backward scales.

We propose instead to simply use the forward scale over the backward scale(s) in these cases. We do so for the following reasons:

1. For these architectures we find empirically that where there is a disparity in ideal forward and backward scales, it is not large.
2. By taking the forward scale, we can ensure strict unit-scale in the forward pass.

The value of the latter point is in terms of what it means for the interpretation of our u-μP multiplier HPs. Consider the $\alpha_{\text{ffn-act}}$ multiplier; with strict unit scale we can say that the standard deviation of activations immediately before this multiplier is 1. Therefore the standard deviation immediately after is $\alpha_{\text{ffn-act}}$. As this multiplier is (by design) the last operation before the ffn activation function, we can say that the interpretation of $\alpha_{\text{ffn-act}}$ is simply to set the input standard deviation to the FFN's activation function. Similar arguments can be made for other u-μP multiplier HPs. This interpretation only holds because we use the forward-scale in our constraints.

## C   THE CHALLENGES WITH μP IN PRACTICE

### C.1   NOT ALL TRAINING SETUPS GIVE μTRANSFER

Lingle (2024) shows that directly applying μP to a decoder LM fails to provide LR transfer across width. Given that the primary use of μP in the literature has been LM training of this kind, this result suggests a significant limitation. How do we reconcile this with the strong LR transfer across width shown for language models in Tensor Programs V?

We answer this in Figure 19. The first training setup (a) is aligned with that used in Tensor Programs V (their Figure 4). There are several atypical aspects to their training setup, primarily the use of a constant LR schedule and a high number of epochs; we outline the precise differences between setup (a) and (b) in Table 8. This overfitting regime makes validation loss unusable, and transfer misleadingly good. When we remove these and shift to a standard Llama training setup (b), optimal HPs begin to drift with width (see Figure 21 for an ablation of individual changes). This confirms Lingle's findings that standard μP is in fact a poor fit for modern LM training. We fix this (c) by the removal of parameters from LayerNorms/RMSNorms, as suggested by Lingle, and the introduction of *independent* weight decay for AdamW, as suggested by Wortsman et al. (2023) [4] (see Wang & Aitchison (2024) for further analysis). With these changes adopted, we recover the strong transfer shown in Tensor Programs V's experiments. Each change is evaluated independently in Figure 20, which shows that the dominant effect is a narrowing of the learning basin due to non-parametric RMSNorms, leading to better learning rate transfer.

### C.2   IT'S NOT CLEAR WHICH HYPERPARAMETERS TO SWEEP

The problem of selecting HPs to sweep can be framed as choosing a subset of the per-tensor $\alpha_W, \sigma_W, \eta_W$ HPs outlined in Section 2.1, and grouping across/within layers. As shown in Table 9, μTransfer experiments in the literature have done this in a variety ways. Practitioners have not

---

[3] As in other work, we use μP as a shorthand for the method outlined in Tensor Programs V, including μTransfer. Strictly speaking, μP ought only to refer to the parametrization outlined in Tensor Programs IV.    [4] Lingle suggests independent weight decay is unstable, but we find it to be more so than Adam or standard AdamW.

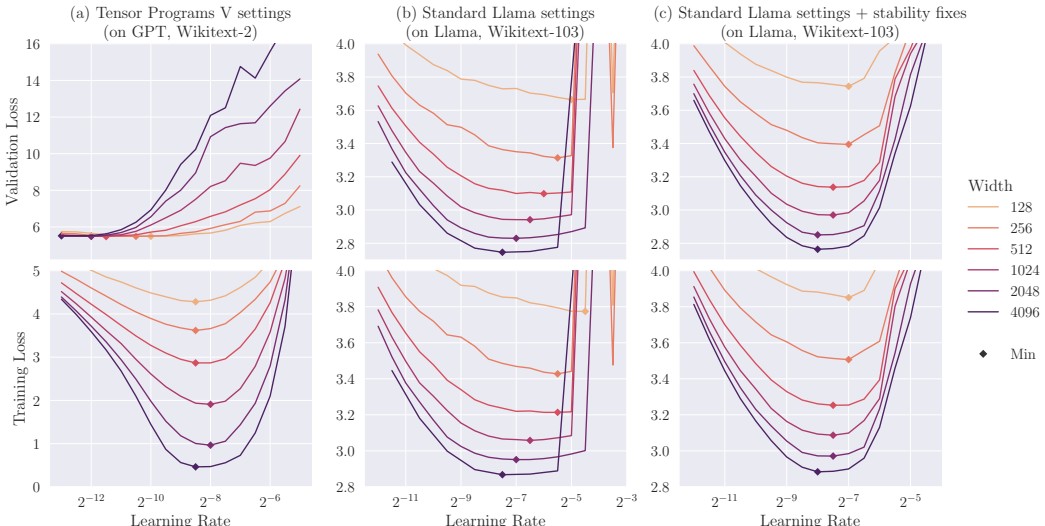

Figure 19: Effective μTransfer does not hold across all training setups. **(a)** We show strong transfer for the unrealistic setup used in Tensor Programs V (too many epochs; constant LR). **(b)** Moving to a more standard Llama training setup, transfer breaks down. **(c)** This is restored by the introduction of two improvements to transfer stability: non-parametric norms and independent weight decay.

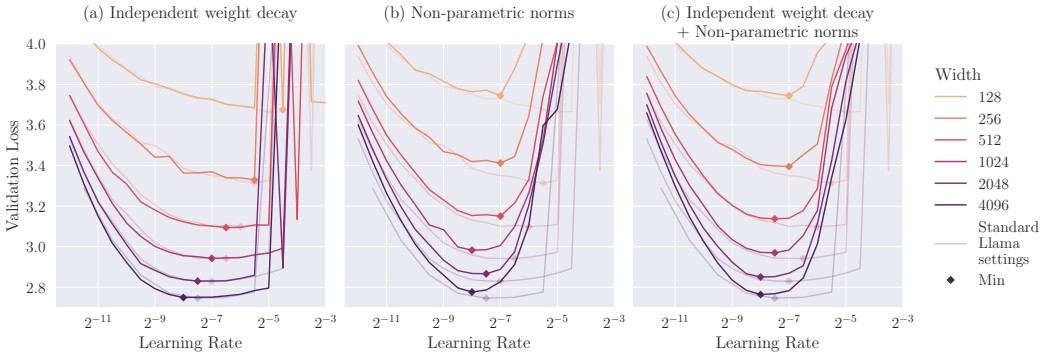

Figure 20: The effect of the individual transfer stability fixes from Figure 19. **(a)** In this setting switching from non-independent to independent weight decay has only a minor effect, though Wortsman et al. (2023) Figure 6 suggests it may be highly valuable in other settings. **(b)** Non-parametric norms give a narrower learning rate basin, leading to better transfer. **(c)** The combination of these, for comparison, matching Figure 19 (c).

| Feature | Tensor Programs V | Standard Llama |
|---|---|---|
| Dataset | wikitext-2 | wikitext-103 |
| Vocab Size | 33278 | 32000 |
| Nsteps | 10000 | 8192 |
| Batch Size | 20 | 64 |
| Optimizer | adam | adamw |
| LR Schedule | constant | cosine |
| Weight Decay | 0 | 0.00012 |
| Positional Encoding | absolute | rotary |
| Norm | layer_norm | rms_norm |
| Dropout | 0.2 | 0 |
| NLayers | 2 | 4 |
| Use Gated FFN | False | True |
| Activation FN | relu | swish |
| FFN Ratio | 4 | 2.75 |
| Final Norm | False | True |
| Base Depth | 1 | 4 |
| Zero Init Readout | True | False |

Table 8: Comparison of Tensor Programs V's standard settings (as best we can tell) and our Standard Llama setup, corresponding to (a) and (b) in Figure 19.

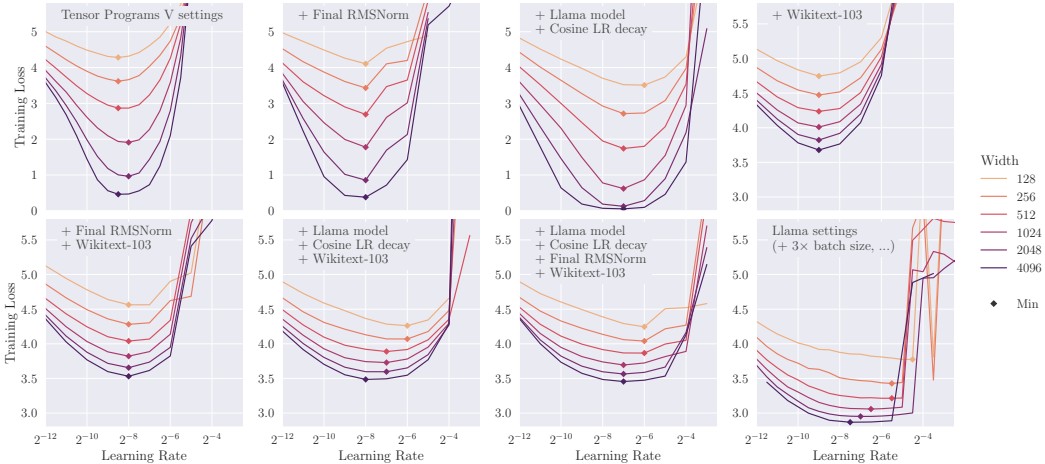

Figure 21: An ablation of the more standard Llama training settings against the Tensor Programs V settings from Figure 19. This shows that the flat basins with poor transfer are not due to a single change, but the combination of a larger dataset (training $< 1$ epoch) and the stronger Llama model are largely responsible. Note that 'Llama model' here indicates a group of changes: rms norm, rotary embeddings & swiglu FFN.

justified these choices, appearing to rely on a mixture of precedent and intuition. We outline two major downsides to the lack of a principled approach.

Firstly, not all groupings of HPs are suitable. Consider the commonly-used global $\sigma_{\text{init}}$ HP. At initialization the activations going into the FFN swish function have $\text{std}(x_{\text{swish}}) \propto \sigma_{W_{\text{gate}}}$, whereas the self-attention softmax activations have $\text{std}(x_{\text{attn}}) \propto \sigma_{W_{\text{Q}}}\sigma_{W_{\text{K}}}$. A global $\sigma$ HP thus has a linear effect on the FFN and a quadratic effect on attention, suggesting that this grouping may be unideal.

Secondly, not all HPs are independent of one another. The key example of this is the interaction between $\sigma_W$ and $\eta_W$. The relative size of a weight update is determined by the ratio $\eta_W/\sigma_W$, not by either HP individually. Because of this, the optimal values for $\sigma$ and $\eta$ depend on each other, which we demonstrate empirically in Section 4.1. This can make the problem of HP search much harder, and may be why hundreds of random-search runs have been required for sweeps in the literature.

## C.3 BASE SHAPE COMPLICATES USAGE

Most practitioners are unlikely to require alignment with an SP model, in which case it is unclear what base-width (and base-depth) should be used. The literature has aligned on a standard base-width of 256 (see Table 9), but this appears to lacking a principled motivation—though the fact that they are not dropped entirely suggests they may be beneficial under u-μP.

Implementing base-shape HPs (see Equation (3)) can also add complications from an engineering perspective. The proposed implementation in the `mup` library (Microsoft, 2024) reflects this, requiring an extra 'base' model to be created and the original model to be re-initialized. This can interact awkwardly with other model-transforms for features like quantization, compilation, etc:

```
import mup

proxy_model = MupModel(d_model=128, ...)      # proxy width
base_model = MupModel(d_model=256, ...)        # base width
mup.set_base_shapes(proxy_model, base_model)   # re-init proxy_model
```

## C.4 μP APPEARS TO STRUGGLE WITH LOW-PRECISION

Finally, we note an interesting contradiction observed in the relationship between μP and low-precision. One of the stated aims for μP is that its activations have $\Theta(1)$-sized coordinates in the limit (Yang et al., 2022, Desiderata J.1). This desideratum is specifically given in order that values can be represented using finite-range floating-point numbers (Yang & Hu, 2021, Section 3). Yet despite numerical stability being central to the theory underlying μP, this is not leveraged to ensure that μP models can *actually* be trained in low-precision. Indeed, for the LLM runs in Tensor Programs V the SP model trains successfully in FP16, while the μP model diverges (attributed to underflow of gradients). We remedy this with u-μP.

# D  A GUIDE TO USING U-μP

We bring together our u-μP scheme presented in Section 3 to form a simple recipe for applying it to a model. The u-μP scheme is designed and validated on a Llama-style architecture, so it may not be applicable or effective on other models, particularly those with substantially different architectures. Exploring this question is an important avenue for future work.

Before applying our scheme, users are encouraged to apply the following pre-requisites to their training setup, based on our analysis of effective μTransfer in Appendix C.1:

- Remove trainable parameters from normalization layers
- Use the *independent* form of AdamW
- Ensure training is in the under-fitting regime (i.e. avoid excessive data repetition)

Having done this, our recipe for using u-μP is as follows:

1. **Replace operations & optimizers with u-μP versions:** Each operation should be replaced by a unit-scaled version (these wrap the existing operations, with added static scales in the forward and backward passes). We have pre-calculated scales for common operations in Appendix B. Parameters should be initialized with unit variance, and Adam(W) adjusted to use the scaling rules defined in Section 3.4 (we refer to the optimizer as Adam in this section, but AdamW should be used if weight decay is required. Other optimizer scaling rules can be determined by the same process we outline).

2. **Choose a set of HPs to sweep:** From the set of HPs outlined in Table 3, select those to be swept. We recommend the extended set, though a basic LR sweep can be effective.

3. **Decide on proxy model config:** The cost of proxy model training should be such that the sweeping process is much less than target model training, while still being as representative as possible. We base our recommendations on the results in Figure 4. In general, width is

the most reliable feature to transfer. Training steps and batch size also give good transfer, so moderate changes here are permissible. Depth is the least reliable feature for transfer, so we only recommend modest changes in depth. We keep the number of warmup steps constant, but always decay to the same final LR when varying the number of steps.

4. **Perform independent HP search:** Following the process outlined in Section 4.1 and Appendix A.5.

5. **Train the target model:** This can be done in FP8 simply by placing casts on matmul inputs (though for our large-scale experiments we found the scales of two operations drifted enough over time that some lightweight dynamic re-scaling was required).

# E  ADDITIONAL BACKGROUND MATERIAL

## E.1  THE MAXIMAL UPDATE PARAMETRIZATION

**Theoretical background**   We do not cover the theory underpinning µP in this paper, presenting only its resulting scaling rules (Table 1). For readers interested in this theory, the extensive Tensor Programs series (Yang, 2019; 2020a; Yang & Littwin, 2021; Yang, 2020b; Yang & Littwin, 2023) builds up a framework from which µP is derived (Yang & Hu, 2021). For those requiring a more accessible introduction, Yang et al. (2023a) show that µP can be derived in a simpler and more general way by placing a spectral scaling condition on the norm of weights and their updates.

**Approaches to HP sweeping in the literature**   Table 9 outlines the ways in which users of µP in the literature have approached HP sweeping. These all follow the approach used in Tensor Programs V of a random sweep, sampling combinations from the joint space of all HPs. The authors of Tensor Programs V note that other more complex methods may be more efficient, but these are considered beyond the scope of their work and have not been used widely. A Bayesian search method was used for the development of MiniCPM (Hu et al., 2024), but the authors give no further details—as they use 400 runs in their sweep it is not clear that this approach makes HP search easier.

Table 9: Sweeping configurations used for a selection of µP models from the literature. The sweeping process is similar across models, the only differences being the choice of discrete or continuous distributions and their ranges. Model references: T.P.V WMT14 (Yang et al., 2022), T.P.V BERT$_{\text{large}}$ (Yang et al., 2022), T.P.V GPT-3 (Yang et al., 2022), MiniCPM (Hu et al., 2024), Cerebras-GPT (Dey et al., 2023a), SµPar (Dey et al., 2024).

| Model | proxy/target tokens used | proxy/target model size | sweep size | base width | HPs swept |
|---|---|---|---|---|---|
| T.P.V WMT14 | 100% | 7.1% | 64 | | $\eta, \alpha_{\text{out}}, \alpha_{\text{attn}}$ |
| T.P.V BERT$_{\text{large}}$ | 10% | 3.7% | 256 | ? | $\eta, \eta_{\text{emb}}, \alpha_{\text{out}}, \alpha_{\text{attn}}, \alpha_{\text{LN}}, \alpha_{\text{bias}}$ |
| T.P.V GPT-3 | 1.3% | 0.6% | 350 | | $\eta, \sigma, \alpha_{\text{emb}}, \alpha_{\text{out}}, \alpha_{\text{attn}}, \alpha_{\text{pos}}$ |
| MiniCPM | 0.008% | 0.45% | 400 | 256 | $\eta, \sigma, \alpha_{\text{emb}}, \alpha_{\text{residual}}$ |
| Cerebras-GPT | 1.1% | 1.5% | 200 | 256 | $\eta, \sigma, \alpha_{\text{emb}}$ |
| SµPar | 6.6% | 6.4% | 350 | 256 | $\eta, \sigma, \alpha_{\text{emb}}$ |

## E.2  UNIT SCALING

**An example: the unit-scaled matmul op**   Here we outline the procedure for calculating the scaling factor of a matmul op, which practitioners can use as a guide for scaling new ops that we do not cover in this paper (see Appendix B).

There are two potential approaches here. The first is to derive scaling factors from an analysis of an op's dynamics. Specifically, given the assumption of unit-scaled inputs, the appropriate scaling factor is the reciprocal of the expected output scale. For a basic matrix-matrix matmul we have,

$$\text{matmul}(X, W) = XW, \qquad X \in \mathbb{R}^{d_{\text{batch}} \times d_{\text{fan-in}}}, \ W \in \mathbb{R}^{d_{\text{fan-in}} \times d_{\text{fan-out}}},$$

where weights and activations are sampled i.i.d. from a centered Gaussian:

$$X_{ij} \sim \mathcal{N}(0, \sigma_X^2),\ W_{jk} \sim \mathcal{N}(0, \sigma_W^2).$$

From this we can derive the expected output scale (i.e. $\sigma(\mathrm{matmul})$):

$$\mathrm{matmul}(X, W)_{ik} = \sum_{j=1}^{d_{\mathrm{fan\text{-}in}}} X_{ij} W_{jk},$$

$$\sigma\left(\mathrm{matmul}(X, W)_{ik}\right) = \sqrt{d_{\mathrm{fan\text{-}in}}}\ \sigma_W\ \sigma_X.$$

Under Unit Scaling we have $\sigma_W = \sigma_X = 1$, and hence the scaling factor required to ensure a unit-scaled output is $1/\sqrt{d_{\mathrm{fan\text{-}in}}}$. This gives our final unit-scaled matmul:

$$\mathrm{u\text{-}matmul}(X, W) = \mathrm{matmul}(X, W)/\sqrt{d_{\mathrm{fan\text{-}in}}}$$

The distributional assumptions made here hold at initialization, but do not over training. A more precise model for the asymptotic behavior of neural networks under training is given by the Tensor Programs framework, but for the purposes of numerics this precise treatment of scale at initialization appears to be sufficient.

The second, less ideal approach to calculating scaling factors is to use experimentation to infer this relationship empirically. In this case, one would sample random initializations and compute the output scale over a range of $d_{\mathrm{fan\text{-}in}}$ values (or whatever HPs one expects the output scale to depend on), fitting a curve to the observed data.

**Applying unit scaling**  To apply Unit Scaling to a model and train in low-precision, the following steps are required:

1. Scale parameter initializations to have zero-mean and unit variance.
2. Replace operations with their unit-scaled equivalents (including and especially the loss, matmuls and residual-adds).
3. *Constrain* the scales of operations which are required to have the same forward and backward factors.
4. Place a simple `.to(fp8)` cast on the inputs to matmuls.

Step 3 relates to the problem of conflicting scales in the forward and backward passes. A single linear layer in a differentiated model requires 3 matmul ops in the forward and backward passes, each requiring a different scaling factor ($\frac{1}{\sqrt{d_{\mathrm{fan\text{-}in}}}}, \frac{1}{\sqrt{d_{\mathrm{fan\text{-}out}}}}, \frac{1}{\sqrt{d_{\mathrm{batch\text{-}size}}}}$). However, using these directly would give invalid gradients. The compromise here is that the activations and activation gradients have their scaling factors *constrained* such that they are equal (the Unit Scaling paper recommends taking the geometric mean; we modify this for u-µP in Appendix B to simply use the forward scale everywhere). Weight gradients can still be given their own scaling factor due to the *cut-edge rule* (as explained in Appendix H).

Step 4 reflects the key benefit of Unit Scaling. Unlike other methods it changes the learning dynamics of a model, but the advantage is that unit-scaled models then 'naturally' generate well-scaled tensors. This means that low-precision arithmetic ideally becomes as simple as placing a cast operation before matmuls as outlined.

## F  UNIT-SCALED PRE-NORM RESIDUAL LAYERS

The popular pre-norm residual network architecture is simple to implement, but problematic to combine with Unit Scaling. It exhibits scale-growth in the skip-stream at initialization, due to the repeated addition of residual connections without subsequent normalization. Here we present a surprising and useful finding: that for any pre-norm model there exists a mathematically-equivalent model where this scale-growth is eliminated, through the careful re-scaling of residual connections.

Note that this section focuses on applying Unit Scaling to *standard* pre-norm models. Only once we have addressed this problem are we able to do the same for u-µP models, as shown in Appendix G.2. Readers only interested in our final u-µP residual implementation may skip ahead to Appendix G.2.2.

### F.1 SCALE GROWTH IN PRE-NORM RESIDUAL NETWORKS

Let's consider a pre-norm residual network of depth $L$:

$$R_0(x) = r_0 x, \tag{4}$$
$$R_l(x) = r_l f_l(R_{l-1}(x)) + R_{l-1}(x), \quad l = 1, .., L \tag{5}$$
$$R_{L+1}(x) = f_{L+1}(R_L(x)) \tag{6}$$

with embedding multiplier $r_0$ and residual branch multipliers $r_l$ for $l = 1, .., L$. To satisfy pre-norm, all $f_l$ are zero-homogeneous functions, i.e. $f_l(\lambda x) = f_l(x)$.

The scale of the skip-stream at initialization as a result of Equation (5) is

$$\sigma(R_l) = \sqrt{r_l^2 \sigma(f_l)^2 + \sigma(R_{l-1})^2} > \sigma(R_{l-1}), \quad l = 1, .., L \tag{7}$$

assuming $r_l^2 \sigma(f_l)^2 > 0$. This shows that scale inevitably grows with the addition of each residual layer.

This scale-growth is clearly incompatible with unit scaling, which aims for $\sigma(R_l) = 1$ for all $l = 0, .., L + 1$. In the following we present an elegant solution to this problem making use of a symmetry transformation available in pre-norm residual architectures.

### F.2 RESIDUAL SYMMETRY IN PRE-NORM ARCHITECTURES

To resolve the problem of scale shift in residual networks demonstrated by Equation (7), we try a slightly more general ansatz:

$$\hat{R}_0(x) = x, \tag{8}$$
$$\hat{R}_l(x) = a_l f_l(\hat{R}_{l-1}(x)) + b_l \hat{R}_{l-1}(x), \tag{9}$$
$$\hat{R}_{L+1}(x) = f_{L+1}(\hat{R}_L(x)) \tag{10}$$

with coefficients $a_l, b_l$. We want to choose these coefficients so that the outputs of $\hat{R}_l$ are unit-scaled if the outputs $f_l, \hat{R}_{l-1}$ are. A similar calculation as in Equation (7) leads to the sufficient condition

$$a_l^2 + b_l^2 = 1, \tag{11}$$

which can be easily satisfied. Having restored Unit Scale, we are faced with another issue. It seems that Equations (8) to (10) describe a different network than Equations (4) to (6), whereas ideally the relation from input to final output should be unchanged when converting the network to Unit Scaling.

Note that the coefficients $a_l, b_l$ are not uniquely defined yet, so our mathematical intuition tells us that we should find an additional constraint to get a unique solution. To find this constraint, let us consider our original residual network in Equations (4) to (6) and analyze how the variance propagates through the network if we assume all the $f_l$ satisfy Unit Scaling and $\sigma(x) = 1$. Let $\sigma_{l-1}^2$ denote the variance of $R_{l-1}$. Then a simple inductive calculation shows that

$$\sigma_{l-1}^2 = \sum_{i=0}^{l-1} r_i^2.$$

By Equation (5) the output of $R_l$ adds a quantity of scale $r_l$ from the residual connection and a quantity of scale $\sigma_{l-1}$ from the skip connection. Intuitively, the *ratio* of these scales should be more important for the overall network dynamics than their absolute values. Thus our constraint becomes preserving the ratio of scales from the original model, through our choice of $a_l, b_l$:

$$\frac{a_l}{b_l} = \frac{\sigma(r_l f_l)}{\sigma_{l-1}} = \frac{r_l}{\sqrt{\sum_{i=0}^{l-1} r_i^2}} =: \tau_l,$$

which, recalling Equation (11), (up to sign) uniquely defines our multipliers $a_l, b_l$ as

$$a_l = \frac{\tau_l}{\sqrt{\tau_l^2 + 1}}, \quad b_l = \frac{1}{\sqrt{\tau_l^2 + 1}} \tag{12}$$

In summary, we propose the modified residual network

$$\hat{R}_0(x) = x, \tag{13}$$

$$\hat{R}_l(x) = \frac{\tau_l}{\sqrt{\tau_l^2 + 1}} f_l(\hat{R}_{l-1}(x)) + \frac{1}{\sqrt{\tau_l^2 + 1}} \hat{R}_{l-1}(x), \tag{14}$$

$$\hat{R}_{L+1}(x) = f_{L+1}(\hat{R}_L(x)), \tag{15}$$

$$\tau_l^2 = \frac{r_l^2}{\sum_{i=0}^{l-1} r_i^2}. \tag{16}$$

Our main result of this section is that this network is indeed mathematically equivalent to the network defined in Equations (4) to (6), under a simple additional structural assumption:

**Lemma F.1.** *Consider $R_l$, $\hat{R}_l$ defined as in Equations (5) and (14) respectively. Then $\hat{R}_l = R_l/\sqrt{\sum_{i=0}^{l} r_i^2}$ for all $l = 0, .., L$.*

Remarkably, this result does not assume the individual network operations $f_l$ actually satisfy Unit Scaling. It is purely a consequence of the pre-norm residual structure. However, only under Unit Scaling can the factors $\tau_l$ be interpreted as the ratio of scales between skip and residual branch.

As a consequence of the lemma, the final residual output $R_L(x)$ is the same as in our original network up to a fixed multiplier. Due to the zero-homogeneity of the final output function $f_{L+1}$ this gives $\hat{R}_{L+1} = f_{L+1}\left(R_L(x)/\sqrt{\sum_{i=0}^{l} r_i^2}\right) = f_{L+1}(R_L(x)) = R_{L+1}$, proving the mathematical equivalence of our residual scheme. Modern LLM architectures like Llama (Touvron et al., 2023a) are pre-norm residual networks of this kind. Hence they admit a faithful unit-scaled reparametrization.

### F.3 PROOF OF LEMMA F.1

*Proof.* This is proved by induction. For the base-case $l = 1$, we have $\tau_1 = r_1/r_0$, giving

$$\hat{R}_1(x) = \frac{\tau_l}{\sqrt{\tau_l^2 + 1}} f_1(x) + \frac{1}{\sqrt{\tau_l^2 + 1}} x$$

$$= (r_1 f_1(x) + r_0 x)/\sqrt{r_0^2 + r_1^2}$$

$$= R_1/\sqrt{r_0^2 + r_1^2}.$$

Then if the statement holds for $l - 1$ we have

$$\hat{R}_l(x) = \frac{\tau_l}{\sqrt{\tau_l^2 + 1}} f_l(\hat{R}_{l-1}(x)) + \frac{1}{\sqrt{\tau_l^2 + 1}} \hat{R}_{l-1}(x)$$

$$= \frac{r_l}{\sqrt{\sum_{i=0}^{l} r_i^2}} f_l(\hat{R}_{l-1}(x)) + \frac{\sqrt{\sum_{i=0}^{l-1} r_i^2}}{\sqrt{\sum_{i=0}^{l} r_i^2}} \hat{R}_{l-1}(x)$$

$$= \left( r_l f_l(\hat{R}_{l-1}(x)) + \sqrt{\sum_{i=0}^{l-1} r_i^2} \hat{R}_{l-1}(x) \right) / \sqrt{\sum_{i=0}^{l} r_i^2}$$

$$= \left( r_l f_l(R_{l-1}(x)) + \sqrt{\sum_{i=0}^{l-1} r_i^2} \frac{R_{l-1}(x)}{\sqrt{\sum_{i=0}^{l-1} r_i^2}} \right) / \sqrt{\sum_{i=0}^{l} r_i^2}$$

$$= (r_l f_l(R_{l-1}(x)) + R_{l-1}(x)) / \sqrt{\sum_{i=0}^{l} r_i^2}$$

$$= R_l(x)/\sqrt{\sum_{i=0}^{l} r_i^2}$$

$\square$

### F.4 UNIT SCALING FOR TRANSFORMER RESIDUALS

The above scheme describes Unit Scaling for arbitrary pre-norm residual networks. We now apply it to the case of pre-norm transformer residual layers.

We can describe a transformer in terms of the residual network given in Equations (4) to (6). Our $f_l$ functions alternate between self-attention layers and feed-forward layers. Implementations differ in the handling of how residual multipliers $r_l$ correspond to HPs. In many cases practitioners simply ignore these $r_l$, but for the sake of expressivity we assume the two types of residual layer each have their own HP, as well as the embedding. In other words,

$$r_l = \begin{cases} \alpha_{\text{emb}} & l = 0 \\ \alpha_{\text{attn-residual}} & l \text{ is odd} \\ \alpha_{\text{ffn-residual}} & l \text{ is even, and } l > 0. \end{cases}$$

To convert this to a Unit Scaled network we apply Equations (13) to (16), from which can derive the following closed-form expression for $\tau_l$:

$$\tau_l^2 = \begin{cases} \dfrac{\alpha_{\text{attn-residual}}^2}{\alpha_{\text{emb}}^2 + \ell\alpha_{\text{attn-residual}}^2 + \ell\alpha_{\text{ffn-residual}}^2} & l \text{ is odd} \\[3ex] \dfrac{\alpha_{\text{ffn-residual}}^2}{\alpha_{\text{emb}}^2 + (\ell+1)\alpha_{\text{attn-residual}}^2 + \ell\alpha_{\text{ffn-residual}}^2} & l \text{ is even.} \end{cases}$$

where $\ell = \lfloor \frac{l-1}{2} \rfloor$.

This gives us a unit-scaled pre-norm residual implementation for a *standard* transformer, which is mathematically equivalent to a non-unit-scaled version. In the next section we augment this by adding in two HPs, in a carefully-designed manner that satisfies our criteria for u-μP HPs, giving us our full residual implementation.

## G JUSTIFYING THE U-μP HYPERPARAMETER SCHEME

Here we justify our particular choice of u-μP HP, as given in Table 3 (with their placement defined in Table 7). We discuss this topic briefly in Section 3.3, stating that all our HPs (excepting the LR) are $\alpha$ HPs, and under u-μP they are now associated with operations instead of weights. All operations have an $\alpha$ HPs, unless they are unary and $k$-homogeneous for $k \geq 0$.

We begin this section by explaining why we apply this rule to the model and how it results in three of our u-μP HPs. We then consider how best to hyperparametrize our residual layers, building on our criteria for HPs given in Section 3.3 and the unit-scaled pre-norm residual scheme in Appendix F.

### G.1 MULTIPLIERS FOR NON-HOMOGENEOUS OPS: $\alpha_{\text{attn-softmax}}$, $\alpha_{\text{ffn-act}}$, $\alpha_{\text{loss-softmax}}$

In this section we derive the rest of our u-μP multipliers. We want to identify the minimal set that can still express all different choices of pre-op scales in the model. The crucial observation is that every pre-scale multiplier $\alpha$ of a unary operation $h \mapsto f(\alpha h)$ can be propagated through the network if $f$ is $k$-homogeneous for some $k > 0$, i.e. $f(\alpha x) = \alpha^k f(x)$, leaving the model and its optimization unchanged. We can iterate this along the computational path until either the next operation is non-homogeneous, non-unary (we are at the end of a residual path), or the next operation is 0-homogeneous (e.g. a norm).

In the first case the accumulated scales are absorbed in the pre-op scale of the non-homogeneous operation (where we introduce a multiplier), in the second case they are absorbed in the residual addition for that branch (where we again introduce a multiplier), and in the final case the scale disappears (so we start over). We now go through the Llama forward computation and follow this paradigm to identify our multipliers in Table 10.

Table 10: A walkthrough of the Llama architecture, showing how our $\alpha_{\text{attn-softmax}}$, $\alpha_{\text{ffn-act}}$ and $\alpha_{\text{loss-softmax}}$ multipliers are derived via an analysis of scale-propagation.

| Op | Scale propagation behavior |
|---|---|
| Embedding | We show in Appendix G.2.1 that the embedding multiplier can be absorbed in the residual multipliers, meaning one is not required here. |
| Attention RMSNorm | This operation is $0$-homogeneous and thus we start over. |
| Query & key projection | Both are linear, meaning their scale is propagated. Multipliers are therefore not required. |
| Query-key matmul | Again linear. As query & key are both generated from the same input, this operation is $2$-homogeneous wrt. that input. Hence it also propagates scale. |
| Softmax | The softmax operation is non-homogeneous. Thus the pre-op scale of the softmax becomes our first multiplier: $\alpha_{\text{attn-softmax}}$. |
| Value | The value layer is linear and hence propagates scale. |
| Softmax-value matmul | Again linear and hence propagates scale. |
| Attention projection | This operation is linear and lies at the end of the attention residual path. Hence there are no more multipliers in the attention block. |
| Residual add | This operation is non-unary and hence receives our second (and third) multipliers: $\alpha_{\text{res}}$, $\alpha_{\text{res-attn-ratio}}$. The manner and motivation for using two multipliers here is justified in the next section. |
| FFN RMSNorm | This operation is $0$-homogeneous and thus we start over. |
| FFN input scale | The input layer is linear, hence it propagates scale. |
| Sigmoid input | This function is non-homogeneous and thus we have our fourth multiplier: $\alpha_{\text{ffn-act}}$. |
| SiLU weight | This layer is also linear and propagates scale. |
| Product | The entry-wise multiplication of the outputs of sigmoid, input layer and SiLU weight is homogeneous and thus propagates scale. |
| FFN output | This layer is linear and at the end of the residual path. Hence there are no more multipliers in the FFN residual block. |
| Residual add | See above. |
| Output RMSNorm | This operation is $0$-homogeneous and thus we start over. |
| Output head | This layer is linear, hence it propagates scale. |
| Loss | The cross-entropy loss is non-homogeneous and leads to our final multiplier: $\alpha_{\text{loss-softmax}}$. |

## G.2 RESIDUAL BRANCH MULTIPLIERS: $\alpha_{\text{res}}$, $\alpha_{\text{res-attn-ratio}}$

In this section we derive our two u-µP residual HPs. We start with the basic, non-unit scaled model we began with in the previous section, outlined in Equations (4) to (6). We described a set of $\alpha_{\text{emb}}, \alpha_{\text{attn-residual}}, \alpha_{\text{ffn-residual}}$ HPs associated with this model in Appendix F.4. However these HPs poorly satisfy our cardinality, independence and interpretability criteria from Section 3.3, so in the Appendix G.2.1 we present a re-parametrization of these HPs designed to better satisfy these points. In Appendix G.2.2 we then combine these HPs with the final unit-scaled pre-norm residual scheme we derived in Appendix F, resulting in our complete u-µP residual scheme.

### G.2.1 IMPROVED HYPERPARAMETERS FOR TRANSFORMER RESIDUALS

To avoid cluttered notation, in this section we rename

$$\alpha_{\text{res}} = \alpha_r, \quad \alpha_{\text{res-attn-ratio}} = \alpha_\rho$$
$$\alpha_{\text{emb}} = \alpha_e, \quad \alpha_{\text{attn-residual}} = \alpha_a \quad \alpha_{\text{ffn-residual}} = \alpha_f.$$

To make the presentation more clear, we derive our new HPs using the standard residual scheme from Equations (4) to (6). For the actual unit scaled implementation one needs to transform the multipliers following Equations (13) to (16), which we do in Section G.2.2.

To facilitate our analysis, we can view the transformer residual output as the sum of three terms:

$$R_L = R_L^{(e)} + R_L^{(a)} + R_L^{(f)},$$
$$R_L^{(e)} := \alpha_e x,$$
$$R_L^{(a)} := \sum_{l=1}^{L/2} \frac{\alpha_a}{\sqrt{L/2}} f_{2l-1}(R_{2l-1}(x)),$$
$$R_L^{(f)} := \sum_{l=1}^{L/2} \frac{\alpha_f}{\sqrt{L/2}} f_{2l}(R_{2l}(x)),$$

and define the average residual scale,

$$\sigma(R_L^{(a,f)})^2 := \frac{\sigma(R_L^{(a)})^2 + \sigma(R_L^{(f)})^2}{2}.$$

Note that we have added in the depth-μP multipliers here, though a similar analysis can be performed for non-depth-μP models. As above, $f_l$ functions alternate between self-attention layers and feed-forward layers.

With respect to our interpretability criterion, we propose two new multipliers that correspond to dynamics in the network which we suggest are important to control at initialization. The first is the ratio of the average scale of the residuals' contributions to those of the embedding, $\alpha_r = \sigma(R_L^{(a,f)})/\sigma(R_L^{(e)})$. The second is the ratio of the scale of the attention-residuals' contributions to those of the feed-forward-residuals, $\alpha_\rho = \sigma(R_L^{(a)})/\sigma(R_L^{(f)})$. Not only do these two ratios control key dynamics of our model, but we can use them to replace our existing $(\alpha_e, \alpha_a, \alpha_f)$ multipliers.

Let us first examine these two quantities for a standard (non-unit-scaled model). Residual functions of the same kind have the same expected output scale at initialization in pre-norm networks, meaning we can denote the output scale $\sigma(f_l(R_l))$ of all self-attention functions as $\sigma_a$, and of all feed-forward functions as $\sigma_f$. We thus have the following scales at the output:

$$\sigma(R_L^{(e)}) = \alpha_e \sigma(x),$$
$$\sigma(R_L^{(a)}) = \frac{\alpha_a}{\sqrt{L/2}} \sigma\left(\sum_{i=1}^{L/2} f_{2l-1}(R_{2l-1})\right) = \alpha_a \sigma_a,$$
$$\sigma(R_L^{(f)}) = \frac{\alpha_f}{\sqrt{L/2}} \sigma\left(\sum_{i=1}^{L/2} f_{2l}(R_{2l})\right) = \alpha_f \sigma_f,$$
$$\sigma(R_L^{(a,f)}) = \sqrt{\frac{(\alpha_a \sigma_a)^2 + (\alpha_f \sigma_f)^2}{2}}.$$

Recalling our definitions of $\alpha_r, \alpha_\rho$ above, this gives us:

$$\alpha_\rho = \frac{\alpha_a}{\alpha_f} \frac{\sigma_a}{\sigma_f},$$

$$\alpha_r = \sqrt{\frac{(\alpha_a \sigma_a)^2 + (\alpha_f \sigma_f)^2}{2 \left(\alpha_e \sigma(x)\right)^2}},$$

$$= \sqrt{\frac{\alpha_\rho^2 + 1}{2}} \frac{\sigma_f}{\sigma(x)} \frac{\alpha_f}{\alpha_e}.$$

The original $\alpha_a, \alpha_f$ multipliers can then be written in terms of $\alpha_r, \alpha_\rho$:

$$\alpha_a = \alpha_\rho \alpha_f \frac{\sigma_f}{\sigma_a}$$

$$\alpha_f = \alpha_r \alpha_e \frac{\sigma(x)}{\sigma_f} \sqrt{\frac{2}{\alpha_\rho^2 + 1}}$$

We have replaced two of the three original multipliers, but still have a dependence on $\alpha_e$ here in our expressions for $\alpha_f$ and $R_L^{(e)}$, which we now remove by dividing it out of our residual branches and embedding. We use the hat $(\hat{\cdot})$ symbol to denote terms that have been divided-through by $\alpha_e$. This new system of equations is equivalent to our old one thanks to the zero-homogeneity of the final post-residual layer:

$$R_{L+1}(x) = f_{L+1}(R_L^{(e)} + R_L^{(a)} + R_L^{(f)})$$

$$= f_{L+1}((R_L^{(e)} + R_L^{(a)} + R_L^{(f)})/\alpha_e)$$

$$= f_{L+1}(\hat{R}_L^{(e)} + \hat{R}_L^{(a)} + \hat{R}_L^{(f)})$$

This gives $\hat{R}_L^{(e)} = \alpha_e x/\alpha_e = x$, removing our first occurrence of $\alpha_e$. Following the division through $\hat{R}_L^{(a)}$ and $\hat{R}_L^{(f)}$, we obtain:

$$\hat{R}_L^{(a)} := \sum_{l=1}^{L/2} \frac{\hat{\alpha}_a}{\sqrt{L/2}} f_{2l-1}(R_{2l-1}),$$

$$\hat{R}_L^{(f)} := \sum_{l=1}^{L/2} \frac{\hat{\alpha}_f}{\sqrt{L/2}} f_{2l}(R_{2l}),$$

$$\hat{\alpha}_a = \alpha_\rho \hat{\alpha}_f \frac{\sigma_f}{\sigma_a},$$

$$\hat{\alpha}_f = \alpha_r \frac{\sigma(x)}{\sigma_f} \sqrt{\frac{2}{\alpha_\rho^2 + 1}}.$$

This system of equations is the same as the original, but with the two $\alpha_e$ terms dropped, meaning our model's multipliers can be expressed in terms of only $\alpha_r$ and $\alpha_\rho$. Using the above equations, any pair of values for $(\alpha_r, \alpha_\rho)$ can be translated back into an equivalent set of values for $(\alpha_e, \alpha_a, \alpha_f)$ such that the output $R_{L+1}(x)$ is the same, meaning that our multipliers are no less expressive than the original set. This satisfies our desired criteria of minimizing the number of multipliers while maintaining expressivity.

We can simplify further in the case of unit-scaled models, which are designed such that $\sigma(x), \sigma_a, \sigma_f$ are all 1 at initialization. In this case our re-parametrization becomes:

$$\hat{\alpha}_a = \alpha_\rho \hat{\alpha}_f, \tag{17}$$

$$\hat{\alpha}_f = \alpha_r \sqrt{\frac{2}{\alpha_\rho^2 + 1}}, \tag{18}$$

$$\hat{\alpha}_e = 1. \tag{19}$$

This is the basis of our claim that Unit Scaling is what enables a more intuitive set of multipliers. Not only do the multipliers $\alpha_r$ and $\alpha_\rho$ represent important dynamics in the network at initialization (the ratio of residual-to-embedding scales, and the ratio of attention-to-feed-forward scales), but it's only via unit scaling that these equations become simple enough to implement in practice. Using equations Equations (17) to (19) for a non-unit scaled network may still be effective, but the interpretation we've given to $\alpha_r$ and $\alpha_\rho$ no longer hold.

Our final desired property is an empirical one: that the most effective choice of one multiplier depends as little as possible on the choice of the other multiplier(s). We demonstrate that our multipliers satisfy this property better than the standard set of residual multipliers in Section 4.1.

### G.2.2 THE FULL U-µP RESIDUAL SCHEME

Here we give the full definition of our u-µP residual scheme, summarizing the results of previous sections. A general pre-norm transformer is implemented as:

$$R_0(x) = c\,x, \tag{20}$$
$$R_l(x) = a_l f_l(R_{l-1}(x)) + b_l R_{l-1}(x), \quad l = 1, .., L \tag{21}$$
$$R_{L+1}(x) = f_{L+1}(R_L(x)), \tag{22}$$

where $a_l, b_l$ and $c$ are scalar multipliers, and the $f_l$ alternate between self-attention and feed-forward layers. We consider our baseline set of µP residual HPs here to be $(\alpha_{\text{emb}}, \alpha_{\text{attn-residual}}, \alpha_{\text{ffn-residual}})$, which we implement (assuming depth-µP branch scaling) as:

$$a_l = \begin{cases} \dfrac{\alpha_{\text{attn-residual}}}{\sqrt{L/2}} & l \text{ is odd (self-attention)} \\[2ex] \dfrac{\alpha_{\text{ffn-residual}}}{\sqrt{L/2}} & l \text{ is even (feed-forward)} \end{cases}$$
$$b_l = 1$$
$$c = \alpha_{\text{emb}}.$$

The corresponding u-µP set of residual HPs is $(\alpha_{\text{res}}, \alpha_{\text{res-attn-ratio}})$, which we implement as:

$$a_l^2 = \frac{\tau_l^2}{\tau_l^2 + 1} \tag{23}$$
$$b_l^2 = \frac{1}{\tau_l^2 + 1} \tag{24}$$
$$c = 1, \tag{25}$$
$$\tag{26}$$

$$\tau_l^2 = \begin{cases} \dfrac{\hat{\alpha}_a^2}{\frac{L}{2} + \ell\hat{\alpha}_a^2 + \ell\hat{\alpha}_f^2} & l \text{ is odd} \\[3ex] \dfrac{\hat{\alpha}_f^2}{\frac{L}{2} + (\ell+1)\hat{\alpha}_a^2 + \ell\hat{\alpha}_f^2} & l \text{ is even} \end{cases}, \quad \ell = \left\lfloor \frac{l-1}{2} \right\rfloor \tag{27}$$

$$\hat{\alpha}_a^2 = \alpha_{\text{res-attn-ratio}}^2\, \hat{\alpha}_f^2 \tag{28}$$
$$\hat{\alpha}_f^2 = \frac{2}{\alpha_{\text{res-attn-ratio}}^2 + 1}\, \alpha_{\text{res}}^2. \tag{29}$$

This is the u-µP residual scheme. It satisfies the three properties that we initially set out to achieve: the variance at initialization of our $R_l(x)$ is always 1, our HPs have a clear and useful interpretation, and our scheme is as expressive as the baseline (which is neither unit-scaled or has interpretable HPs).

## H  THE CUT-EDGE RULE

In the section we review the notion of *constraints* used for scaling operations in a computational graph. For a more thorough, generalized treatment, please refer to Section 5.1 and Appendix E.4 of the Unit Scaling paper (Blake et al., 2023).

For simplicity, we will only discuss the cut-edge rule in the context of a typical neural network. For each operation $f$, parametrized by $\theta$ taking input $x$ and emitting output $y$, a user must choose how to scale $y$, $\nabla_x$ and $\nabla_\theta$ (gradient of loss w.r.t. $x$ and $\theta$ respectively). In the simplest case, where there are no further data dependencies, we can simply choose factors that preserve unit scale. In more complex scenarios, we must balance the need for each tensor to be unit-scaled and for gradients to be correct up to a constant factor.

In particular, a problem emerges in the presence of residual blocks in which $y = x + f(x; \theta)$. In these circumstances, $\nabla_x$ is computed as the sum of residual gradient $\nabla_f \frac{\partial f}{\partial x}$ and skip gradient $\nabla_y$. If we choose not to insert scaling factors into our graph, $\nabla_f \frac{\partial f}{\partial x}$ and $\nabla_y$ will have some ratio of scale $r$. However, if we have chosen to rescale the gradient of operations in $f$, then $\nabla_f \frac{\partial f}{\partial x}$ will have been rescaled by some $s$. This means the new ratio of $\nabla_f \frac{\partial f}{\partial x}$ and $\nabla_y$ will be $r \cdot s$. Therefore, when adding these together, $\nabla_x$ is no longer a correct gradient up to a constant factor.

How do you remedy this? If we can ensure that the scaling factors are the same for both the input gradients and outputs of an op, we will have $s = 1$. This ensures that gradients for inputs to residual blocks are correct up to a constant factor.

How do you decide when you are free to preserve unit scale, and when to constrain scaling factors to be the same? We previously define the *cut-edge rule* (Blake et al., 2023) for computational graphs where nodes represent forward pass operations and edges represent operation outputs. If an input edge is a *cut-edge*, i.e., the number of connected components in the graph would increase upon deletion (examples in a typical transformer model: output of embedding gather, output of a residual add, output of final norm, output token logits, weights), there is no need to constrain the scales of the operation's output edge and the input edge gradient. For all other input edges (e.g., inputs to a residual add, intermediates computed along a residual branch), the scales of gradients and outputs should be constrained.

## I FROM µP TO U-µP

Here we outline additional details to help readers follow the process of deriving u-µP from the combination of Unit Scaling and µP. Our first step of dropping $\sigma_W$ and base-fan-in, and moving $\alpha_W$s to functions, results in Table 11. This intermediate scheme does not yet satisfy Unit Scaling, but simplifies the HP rules in preparation for further changes. Note that we have also removed $\hat{\eta}_{\text{emb}}$ as we don't include this HP in our u-µP extended HP set. We have included residual scaling rules here, in accordance with depth-µP, which we intend u-µP to satisfy, though our standard µP implementation doesn't use it.

Table 11: An intermediate scheme resulting from dropping those HPs from µP which are not needed under u-µP.

| ABC-multiplier | | Weight Type | | | Residual |
|---|---|---|---|---|---|
| | | Input | Hidden | Output | |
| parameter | $(A_W)$ | 1 | 1 | $\frac{1}{\text{fan-in}}$ | $\frac{1}{\sqrt{\text{depth}}}$ * |
| initialization | $(B_W)$ | 1 | $\frac{1}{\sqrt{\text{fan-in}}}$ | 1 | — |
| Adam LR | $(C_W)$ | $\eta$ | $\eta \frac{1}{\text{fan-in}}$ | $\eta$ | $\frac{1}{\sqrt{\text{depth}}}$ |

## J LOW-PRECISION AND ITS TRADE-OFFS

**Number formats for deep learning** The standard numerical representations used in deep learning are the set of formats defined by the IEEE 754 floating-point standard (IEEE Computer Society, 2019). IEEE floats comprise three elements: a sign bit, exponent bits, and mantissa bits. The number of exponent bits determines the *range* of a format, while the mantissa determines the *precision*[5].

We refer readers to Blake et al. (2023), Section 3.1 for a comprehensive overview of floating-point representations.

The default format used for training is the single-precision floating-point format, commonly known as FP32, with some hardware providers automatically casting it to the smaller TF32 compute mode for accelerated arithmetic. The 16-bit FP16 and BF16 formats were later introduced, and more recently the FP8 E5 & E4 formats (Sun et al., 2019; Noune et al., 2022; Micikevicius et al., 2022). The higher range of E5 has typically been used for gradients, while the higher precision of E4 has been seen as necessary for weights and activations. Other aspects of training such as the optimizer state and cross-device communication have also been put into FP8 (Peng et al., 2023), though not all tensors are amenable to being run in the lowest precision (Dettmers et al., 2022) without degradation. The use of multiple formats is known as *mixed precision* (Micikevicius et al., 2018). A comparison of these formats is given in Table 12.

Table 12: A comparison of deep learning formats. E indicates exponent bits, and M mantissa bits. The smaller formats typically give more FLOPS, at the expense of reduced range and/or precision.

| Format | E | M | \| max \| | \| min normal \| | \| min subnormal \| | FLOPS (vs TF32) |
|--------|---|---|-----------|------------------|---------------------|-----------------|
| FP32   | 8 | 23 | $3.4 \times 10^{38}$ | $1.2 \times 10^{-38}$ | $1.4 \times 10^{-45}$ | $< 1\times$ |
| TF32   | 8 | 10 | $3.4 \times 10^{38}$ | $1.2 \times 10^{-38}$ | $1.1 \times 10^{-41}$ | $1\times$ |
| BF16   | 8 | 7 | $3.4 \times 10^{38}$ | $1.2 \times 10^{-38}$ | $9.2 \times 10^{-41}$ | $2\times$ |
| FP16   | 5 | 10 | 65504 | $6.1 \times 10^{-5}$ | $6.0 \times 10^{-8}$ | $2\times$ |
| FP8 E5 | 5 | 2 | 57344 | $6.1 \times 10^{-5}$ | $1.5 \times 10^{-5}$ | $4\times$ |
| FP8 E4 | 4 | 3 | 448 | $1.6 \times 10^{-2}$ | $2.0 \times 10^{-3}$ | $4\times$ |

**The benefits of low-precision**    Using numerical representations with fewer bits facilitates the design of more efficient arithmetic in hardware, typically leading to a linear increase in peak FLOPS (as shown in Table 12). As large-scale training efforts are typically compute-bound due to the size of matmuls (Narayanan et al., 2021), putting the inputs to these operations in low-precision formats has a substantial impact on training efficiency. Low-precision formats also reduce the other two common performance constraints: for memory-bandwidth-bound models they require fewer bits to be transmitted, and for memory-size-bound models they require fewer bits to be stored.

**The challenges of low-precision**    Unfortunately, moving to low-precision formats also increases *quantization error*. For values within the representable range this takes the form of *rounding error*, and for values outside it, *clipping error* (both overflow and underflow). Rounding error tends to be an intrinsic problem: the number of mantissa bits dictates the expected accuracy of representations and this cannot easily be changed. In contrast, clipping error is often eliminated by scaling a tensor so that its values lie within the range of a format. Note that a multiplicative change in values of this kind doesn't affect the (relative) rounding error, due to the exponential spacing of values. Most research into making low-precision work has focused on the problem of scaling tensors in this way.

Simply casting all tensors to FP16 or FP8 tends to impair training, largely due to clipping error. For FP16, this primarily affects gradients. Micikevicius et al. (2018) address this by introducing a fixed global *loss-scale* HP, which multiplies the loss value in the backward pass, artificially up-scaling gradients to lie within FP16 range. *Automatic loss scaling* (Kuchaiev et al., 2018) builds upon this idea, making the loss-scale a dynamic value that is tuned during training.

The later BF16 format has the same range as FP32, making loss scaling unnecessary. For FP8 no such range-equivalent format can exist, so the problem of clipping error must be addressed. Most FP8 implementations have done so by moving from a global loss-scale to a local scale for each FP8 tensor. In pseudo-code, this takes the form:

```
a = scale(A)
b = scale(B)
A = to_fp8(A / a)
```

---

[5] Confusingly, the term *low-precision* tends to indicate using <32 bit-width formats, so in this context *precision* also reflects the number of exponent bits as well as the usual mantissa bits.

```
B = to_fp8(B / b)
C = (a * b) * matmul(A, B)
```

where we assume that `matmul` takes inputs in FP8 and directly produces the output in higher precision.

The result of the `scale()` operation can either be a fixed scale determined before training (Noune et al., 2022), or in the case of Transformer Engine (NVIDIA, 2024a), computed dynamically as a function of the 'absmax' of the input tensor (though they introduce a delay across time-steps, to facilitate an efficient fused kernel). Increasing granularity and computing scales dynamically using this kind of method inevitably adds complexity (from both a logical and implementation perspective), as well the potential for computational overhead. Unit Scaling generally avoids the need for matmul input scaling.

## K   BENCHMARKING SCALED MATRIX MULTIPLICATION IMPLEMENTATION IN PYTORCH

Given that the end-goal of leveraging u-mup's low-precision properties is to speed up training and reduce memory usage, it's reasonable to ask why we don't investigate this experimentally. The answer relates to the relative immaturity of the FP8 training software stack - a lack of open, efficient FP8 kernels for compute and communication mean significant additional engineering effort is required to attain expected speedups over the full model.

Here we show that u-μP's static scaling factors add no overhead to matmuls in FP8, and hence ought to be able to reach close to the maximal FP8 throughput attainable for the full model.

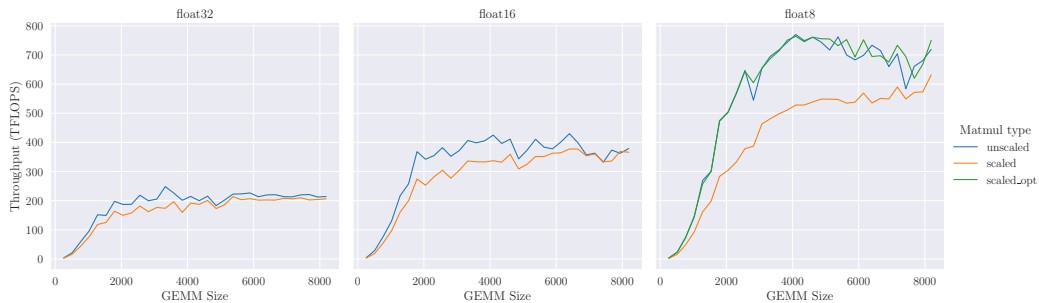

Figure 22: Square matrix multiplication throughput in TFLOPs with and without scaling factors applied to the output across 32-, 16-, and 8-bit float dtypes on NVIDIA H100 PCIe. Naive implementation in PyTorch.

Standard strategies for FP8 training require expensive statistics gathering (e.g., amax) per tensor. A key benefit of u-μP for FP8 training is that it instead provides us with static scaling factors to rescale operation outputs. Even a naive implementation in pytorch can achieve a minimal drop in hardware utilization.

Figure 22 demonstrates hardware utilization for FP8, FP16, and FP32 matrix multiplications on a single NVIDIA H100 PCIe card. For FP16 and FP32, `torch.matmul` is used, whereas `torch._scaled_mm` is used for FP8. Comparing "scaled" to "unscaled" matrix multiplication demonstrates a 30%, 20%, and 10% drop in hardware utilization for each data type respectively. In the case of FP8, where the drop in utilization is most pronounced, utilization can be recovered by passing the scaling factor as a scale associated with one of the two input tensors.

It should be noted that as of PyTorch version 2.3, `torch._scaled_mm` always computes amax as well as the matrix multiplication. The performance of FP8 matrix multiplications could be higher without this overhead.

The above analysis focuses on throughput; significant memory savings are also possible through the use of FP8, though how this affects the total memory footprint depends on various additional variables and the overall distributed training setup. The following factors are play a significant role: typically the main memory bottlenecks are the optimizer states, which are kept in full precision. This footprint can be reduced by applying ZeRO sharding (Rajbhandari et al., 2020), though for significant gains the number of data parallel processes needs to be sufficiently large and ZeRO stage 2 or 3 are required. In these settings the memory footprint of activations and gradients becomes significant, and quantizing these to lower precision promises further memory savings, though may be non-trivial (Peng et al., 2023).

## L  ATTENTION OUTPUT RMS GROWS WITH MODEL DEPTH

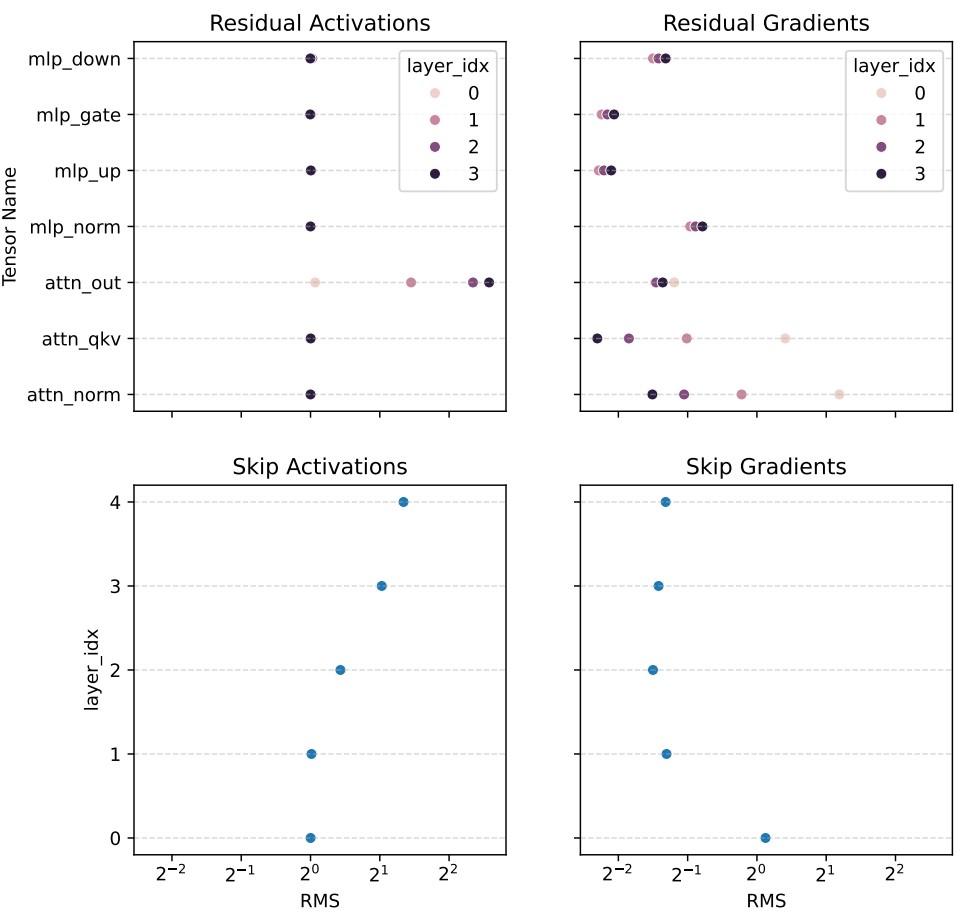

Figure 23: Scale of intermediate tensors grows with depth at initialization. Top left: Intermediate activation tensor RMS along the residual branch. Only the attention outputs after the first layer are not unit-scaled. Bottom left: Skip activation tensor RMS. Scale growth in attention outputs drives growth in skip activation scales. Note that `layer_idx`= 0 corresponds to the embedding output, and `layer_idx`= 4 corresponds to the final layer outputs. Top right: Intermediate gradient tensor RMS along the residual branch. Growth in the attention output scale drives growth in attention qkv gradient scales. Bottom Right: Skip gradient tensor RMS. The scale of output activations induces a global rescaling of the gradients.

A core assumption in deriving per-op scaling factors is that each input to an operation has zero mean, unit-variance, and uncorrelated elements at initialization. This is trivially true for weights and by extension the token embeddings taken as input to the transformer trunk. However, this is not guaranteed for intermediate results and gradients if an operation in the computational graph induces correlation in the elements. In such a scenario our scaling factors will not return unit-variance outputs as we will not have corrected for these correlations in the inputs. As we then increase the depth of the network, where the same operation is left to amplify correlations, we can end up with variance in intermediate results and gradients scaling with depth

Figure 23 illustrates this phenomenon in a unit-scaled four-layer Llama model with width=256. All activation tensors in the residual branches are unit-scaled, except for the output of the attention layers. We also see that the variance of attention outputs grows with depth. Since Llama models use pre-norm on the residual-branch, residual-branch inputs will revert to unit-scale again until they reach another instance of the correlation-inducing operation. As we add under-scaled attention layer results back to the skip-branch, our skip tensor variances grow with depth as our residual-add assumes unit-variance inputs. This has a knock-on effect on the global scaling of the gradients since the Jacobian of the final norm will scale the gradient by the inverse of the final skip tensor variance.

So which operation induces correlation in the attention output at initialization? For the default case where all multipliers are set to 1, our $1/d$ scaling of attention logits results in a sufficiently high temperature that attention probabilities are effectively uniform. With causal masking, we effectively take a running mean across the value tensor along the sequence dimension. As a result, each subsequent token representation is correlated with the last. Since we derive appropriate scaling factors for the first layer, we do not see scale growth emerging until the second layer, where correlations accumulate during the next effective running mean.

We leave it to future work to offer a solution to scale growth created by correlation in intermediate tensors. We note that this is scale growth emergent at initialization, but we also see scale growth in other intermediate tensors during training. Whether scale growth during training is related to the phenomenon outlined here remains to be seen.

