# OpenReview forum: "u-$\mu$P: The Unit-Scaled Maximal Update Parametrization"
_ICLR.cc/2025/Conference — ICLR 2025 Spotlight_

### Official Review · Reviewer_uAPB · 2024-10-28

**Soundness:** 3
**Presentation:** 3
**Contribution:** 3
**Rating:** 8
**Confidence:** 3

**Summary:**

This work proposed u-$\mu P$, which combines $\mu P$ with unit scaling to facilitate hyper-parameters (HPs) search. u-$\mu P$ does not require a base shape and the HPs have less interdependency. Moreover,  u-$\mu P$ empirically maintains unit scaling for the activations, weights and gradients, which enables FP8 low-precision training.

**Strengths:**

The paper is well-written and the idea is clearly delivered. The authors have conducted extensive experiments to compare the performance of the proposed methods with the original $\mu P$.

**Weaknesses:**

* As the authors discussed, this work lacks a comparison with other proposed methods (e.g., Large et al., 2024).
* There is no theoretical justification of why choosing a different scaling for the embedding learning rate.

**Questions:**

* Maybe the authors already mentioned this, but I wonder how is the performance of u-$\mu P$ with embedding LR =1 compared with $\mu P$?

* I am curious about how whether u-$\mu P$ can be applied to non-transformer models and how to select the HPs (as in line 304-316) in this case? I am not asking for additional experiments, any useful insights or discussion would be appreciated.

* Typo in Figure2 right panel: the solid and dotted lines are swapped?

---

> ### Author Response · Authors · 2024-11-23
>
> Thank you very much for your review of our work, and encouraging feedback. On your questions and concerns:
>
> **"How is the performance of u-mup with embedding LR =1 compared with mup"**
>
> The parametrization used in u-mup with embedding LR =1 is equivalent to regular mup under abc-symmetry, meaning performance should be the same. Everett et al. (2024) show this result explicitly in terms of their equivalence classes, with Mean Field Parametrization (which is equivalent to u-mup's scheme with embedding LR =1) belonging to the same "equivalence class" as mup. They show that this equivalence can break if the ratio of gradient size to Adam epsilon becomes too small, but as we up-scale our gradients we avoid this scenario (see also our response to reviewer CvCm on this topic).
>
> Another key part of u-µP is our improved HP scheme. Applying this to mup would be very interesting additional work. Our HP scheme largely makes sense in the context of u-µP, as it's designed to maintain unit-scale wherever possible, so adding it to regular mup would somewhat undermine its intended purpose (hence why we didn't run the experiment), but empirically it would be interesting to explore this anyway, to perhaps separate the effects of the parametrization and HP scheme. We have prioritized other, simpler experiments for this rebuttal period, but hope we or others will examine this idea in the future.
>
> **"Typo in Figure2 right panel: the solid and dotted lines are swapped?"**
>
> Yes they are, thank you for pointing this out. We have corrected this in the paper.
>
> **"This work lacks a comparison with other proposed methods..."**
>
> Though we discuss other methods in our related work — including Modula by Large at al., 2024, and the parametrization study by Everett et al., 2024 — we agree that further analysis of u-µP in light of these methods would be valuable for assessing the merits of different approaches. In both of these cases many aspects are orthogonal to u-µP: one could certainly conceive of other unit-scaled parametrizations, including a version of Modula that admits unit-scaling.
>
> Properly assessing the 8 schemes outlined in Everett (of which we cover ~3) would require a large number of training runs, and deriving a unit-scaled Modula appears sufficiently complex to warrant a separate project. We have updated our related work section to make this point explicitly in the paper. Given the wide scope of our existing experiments we will leave this comparison to follow-up work, but hope that others will explore u-MFP, u-NTK u-Modula, etc.
>
> **"I am curious about how whether u-mup can be applied to non-transformer models"**
>
> Like the original mup and Unit Scaling methods, u-mup can be applied to a wide range of architectures. For the HP selection we focused on Transformer models in this work, but the general recipe of moving weight multipliers into non-homogeneous operations works for pretty much any architecture. See Appendix G.1, G.2 and Table 9. Based on your feedback that this was not sufficiently clear, we have added two remarks about this in the main text.
>
> **"No theoretical justification for the embedding learning rate scaling"**
>
> We would certainly like to understand this better, but deriving a theoretical explanation for our empirically-justified rule appears non-trivial. Reviewer CvCm makes a similar point, which we have responded to under the heading "The learning rate in the embedding seems unnatural and contradicts the original mup paper" - we refer you to this response for more detail, including a common precedent for this kind of embedding under-scaling in standard training.
>
> ---
>
> **Summary**
>
> We hope that our explanations here and that the tweaks we've made to the paper satisfy the questions you raised in your review. We thank you for raising these points and appreciate your positive feedback.

---

> > ### Comment · Reviewer_uAPB · 2024-11-26
> >
> > Thanks for the responses! My questions are answered and I have increased my confidence score.

---

### Official Review · Reviewer_CvCm · 2024-11-01

**Soundness:** 2
**Presentation:** 3
**Contribution:** 3
**Rating:** 6
**Confidence:** 4

**Summary:**

This paper combines two approaches—(1) mup and (2) unit-scale—to enable hyperparameter transfer for low-precision training (FP8). The paper is overall well-written and contains many details and ablations.

I do like this direction of research, and low-precision training is important for speeding up training, lowering training costs, and enabling more research.

However, I have several concerns regarding the paper listed below. I am happy to raise the score if these are addressed properly.



---- update ----

I am delighted with the promised change. in particular, down play the batch, depth scaling results; detailed discussion of embedding scaling and potential limitation;

i raised my score to 6 ( i would give 7 if that option is available).

**Strengths:**

I think this is important research direction that has many practical values. The paper contains a lot of useful details (many of them are in  the appendix). I really appreciate that.

The embedding scaling rule seems interesting and novel but also controversal.

**Weaknesses:**

- Transfer across batch, depth, training steps is not convincing (Fig. 4).
 - (important) The learning rate in the embedding seems unnatural and contradicts the original mup paper. In the infinite-width setting, the update will go to zero, and the input layer is frozen; this doesn't seem right to me.
 - (Important) Everett also studies hyperparameter transfer thoroughly and is highly related to this paper. Please make a more comprehensive comparison. In particular, the mean-field parameterization is very close to the unit-scale proposed here. Please clarify what's new and what the differences are.
- Citations to previous mean-field papers are needed and should have been discussed.
- The results of the paper seem to contradict some results of the original mup paper. I would love to see a table/section summarizing them + an explanation of why the original mup setup is not right.
- I  also want to see a wall clock time comparison between fp-8 vs. bf-16 runs at different scales, which is why we want to use fp-8. In addition, it may be good to know how much memory can be saved.

Again, I think this can be a good paper and I would like it to get improved.

**Questions:**

Independent weight decay vs coupled weight decay? need more clarification. I am not sure which one to use. Lingle proposes to use wd=0.1 * LR, while og mup, wortsman and here propose scale-independent weight decay.

Fig 2. Are the legends regarding C_embed correct? It says C_embed = 1 is better than C_embed  = 1/ root(fan_out).

The uses of non-trainable rmsnorm "scales" seem non-standard to me, have you done ablations by yourself?

---

> ### Author Response · Authors · 2024-11-15
>
> Thanks for providing us with feedback, we appreciate the care taken when reading the paper and writing your response. We wanted to clarify one point to make sure we address the intent of your question:
>
> Regarding your comment "the results of the paper seem to contradict some results of the original mup paper" - did you have something particular in mind? There are several points where our message differs a little from that of the original mup paper: the changed embedding LR scaling rule, our argument for the need for independent weight decay \& non-parametric norms, our results showing worse transfer results for regular mup than suggested by the original paper.
>
> Was your suggestion about us summarizing all of these points, or just e.g. the transfer results discrepancy?

---

> > ### Comment · Reviewer_CvCm · 2024-11-15
> > **re**
> >
> > it is more the transfer discrepancy;
> > embedding lr is one key part;
> > in addition, the og mup paper seems to have cleaner transfer for depth (fig 4 in your paper vs fig 4 in mup paper), batch size (fig 4 your paper vs fig 19 mup paper) than what is here. ( # training steps drifts a little);

---

> ### Author Response · Authors · 2024-11-23
>
> Thank you for the careful and detailed review of our work. To respond to your questions:
>
> **"Transfer across batch, depth, training steps is not convincing"**
>
> We are also not convinced that these axes give strong transfer! We overstated this in the paper originally by saying that they give "approximately constant" optima, which particularly for depth is hard to justify.
>
> Our language in the paper has been changed to reflect this, now stating that "there is a small drift for training steps and batch size, and a larger one with depth. Hence we recommend proxy models which primarily differ in width, moderately in steps and batch size, and least in depth."
>
> We emphasize that our purpose in showing these results was not to argue for strong transfer across these axes, but simply to show practitioners how well LR transfers for steps, batch size and depth, for practical purposes. We have also changed the text to emphasize that these drifting optima are similar for u-mup and mup (i.e. we don't degrade it), comparing Figure 4 with Figure 13 — as expected, given our changes are not designed to modify behavior over these axes.
>
> **"The learning rate in the embedding seems unnatural and contradicts the original mup paper"**
>
> Our learning rate rule is indeed a major deviation from standard mup, effectively freezing the embeddings at very large width. We are well aware of this and were surprised by the results of our experiments, yet we felt it best to adapt our scheme to follow this clear empirical trend. We do not yet have an explanation for our embedding learning rate (our preliminary attempts have been without success; the answer would appear to be non-trivial), which is why we encourage further investigation as follow-up work.
>
> We understand the reviewer's instinct that this doesn't seem right (e.g. indicating an error on our behalf). We only became comfortable with this rule upon realizing that the way in which most practitioners train—without mup and with a global LR—also induces this kind of under-scaling of embedding gradients. To see this, consider the Standard Parametrization, as outlined in Table 1 of Everett et al. (2024). This table assumes per-layer-type LRs, but many practitioners use a single LR and scale it using some heuristic, likely between 1/sqrt(n) and 1/n, which are the no-alignment and full-alignment LR scales for hidden weights. Doing so with unit-initialized embeddings leads to frozen parameters in the infinite-width limit, just as in our case.
>
> We don't claim that the above observations justify our LR rule, but we think there is some indication that the role of the word embedding may be complex and needs to be understood better. We hope that follow-up work is able to explain this phenomenon, and that in posing it we provide a valuable problem for the research community to solve.
>
> **"Please make a more comprehensive comparison to Everett"**
>
> The main reason we didn't do this was a matter of timing - Everett's work came out after much of our theoretical analysis was done (and is concurrent under the ICLR definition). Having said this, we consider the comparison very valuable so have now added text to our related work discussing this.
>
> In it we outline that our u-µP parametrization is equivalent to the mean field parametrization with the "full alignment" assumption from Everett et al, plus our modified embedding LR rule. Our discussion of abc-symmetry is also paralleled by their notion of "equivalence classes" of parametrizations. They also show that under mean field the size of the gradient becomes small enough that the epsilon term in Adam cannot be ignored and hence breaks this equivalence, which they correct by scaling down the epsilon (or using a new optimizer). Interestingly we now realize that unit scaling solves this problem by doing the inverse: scaling up the gradient (to unit-scale) to make it larger compared with epsilon. In a sense Unit Scaling is naturally robust to this kind of problem because, unlike other parametrization approaches, it mandates a fixed scale of gradients.
>
> We also think that the family of parametrizations defined by Everett are interesting targets for Unit Scaling, and are keen to apply our numerical and HP approaches to these other parametrizations.
>
> **"Citations to previous mean-field papers are needed"**
>
> Thank you for pointing this out — we've added appropriate citations and a short discussion to our related work section, as well as mentioning this topic in our newly-added discussion on Everett et al. above.

---

> > ### Author Response · Authors · 2024-11-23
> >
> > **"I would love to see a table/section summarizing [contradictions] + an explanation of why the original mup setup is not right"**
> >
> > We agree that our mup baseline results differ from the Tensor Programs V paper, and hope to provide an improved explanation. We focus first on width transfer as this is the key theoretical property emerging from mup. Recalling the Tensor Programs V vs Standard Llama plots in Figures 19 (a) and (b), we've run a series of ablations to explain the changes in loss curves as a result of these different regimes. These ablations can be seen in Figure 21, along with a new Table 8 containing all the changes we made to the setup from Tensor Programs V to reach "Standard Llama" training.
> >
> > There is no single change that explains the degradation (optimal LR drift and steep "cliffs") of mup on Standard Llama, and the picture appears complex. The most significant changes appear to be: 1) leaving the over-fitting regime by using a larger dataset (they appear to use 15-16 epochs of WikiText-2 leading to overfitting), and 2) using the stronger Llama-style architecture, which adds RMSNorm, Gated FFN and relative positional encodings. Together, these changes have the effect of flattening the LR optimum basin and introducing steep "cliffs" at high learning rate. (note: our 4k-width results for these plots are in progress)
> >
> > As for batch size and depth, we have not been able to run a similar ablation, but an inspection of the differences in Table 8 provides some insight. For their batch size results, the fact that they are strongly in the over-fitting regime may help stabilize their results, though this is speculation.
> >
> > We do not have an explanation for their surprisingly stable depth plot, which we have not observed in any of our experiments. We note that the depth transfer shown for standard (i.e. non-depth) mup in Tensor Programs VI (e.g. Figure 10, middle) is far less stable than the standard mup results in Tensor Programs V, despite TPV not appearing to have any of the depth-scaling features, suggesting some inconsistency in these results. Indeed, the depth results for standard mup in TPV are stronger than those for depth-mup in TPVI (e.g. Figure 16). Our depth-mup results are not dissimilar to those in TPVI though, which suggests there is not a major disparity in depth-scaling for u-mup.
> >
> > **"wall clock time comparison between fp-8 vs. bf-16 runs at different scales"**
> >
> > Given that the end-goal of leveraging u-mup's low-precision properties is to speed up training and reduce memory usage, it's reasonable to ask why we don't investigate this experimentally. The answer relates to the relative immaturity of the FP8 training software stack - a lack of open, efficient FP8 kernels for compute and communication mean that we don't see the gains in FP8 that are potentially available, and that other closed-source efforts have demonstrated (e.g. 1.5x speedup from https://www.databricks.com/blog/turbocharged-training-optimizing-databricks-mosaic-ai-stack-fp8).
> >
> > As for potential memory savings, with 70 percent of our linear layers in FP8, we save about 35 percent of memory for the transformer weights. How this affects the total memory footprint depends on various additional variables and the overall distributed training setup. Because of this we cannot give a simple estimate for how much memory can be saved during training, but the following factors are play a significant role: typically the main memory bottlenecks are the optimizer states, which are kept in full precision. This footprint can be reduced by applying ZeRO sharding, though for significant gains the number of data parallel processes needs to be sufficiently large and ZeRO stage 2 or 3 are required. In these settings the memory footprint of activations and gradients becomes significant, and quantizing these to lower precision promises further memory savings, though may be non-trivial (see Peng et al., 2023).
> >
> > As this is primarily a research project, we leave optimizing FP8 performance to future work. It is enough for us to show that our method *ought* to be able to achieve these efficiencies in theory, with the same engineering effort. Showing this for throughput is the purpose of our micro-benchmarking in Appendix K: we show that the additional op we introduce into linear layers (a matrix-scalar multiplication) can be fused into a matmul with no noticeable overhead in FP8 using torch.compile (similar results ought to be realizable with scaling factors applied to other ops). Based on this, we reason that we should be able to reach an MFU equal to any other FP8 training effort as our scheme is otherwise a standard model, and potentially better if they are using a more complex (e.g. dynamic) scaling approach.
> >
> > We have added text to Appendix K making the above arguments, as we feel it's valuable context for understanding the implications of u-mup for low-precision efficiency.

---

> > > ### Author Response · Authors · 2024-11-23
> > >
> > > **"Independent weight decay vs coupled weight decay? need more clarification"**
> > >
> > > We've run an extra experiment to investigate this question (Figure 20). It appears that in our setting independent weight decay performs similarly to coupled weight decay. However, in in Wortsman et al. (their Fig. 6), non-independent decay appears to degrade faster at high LRs than in our setting, which we believe is due to a higher effective level of weight decay in their experiments. This supports the use of the non-independent form, and in our experiments we see no detrimental effects; hence we continue to recommend it. We don't dismiss the concern raised by Lingle in stating that it can lead to training instabilities for large models, but others including us have not observed this phenomenon and Lingle does not provide the experimental results which would assist in investigating it.
> > >
> > > **"Fig 2. Are the legends regarding C_embed correct?"**
> > >
> > > Thanks for pointing this out. We've now fixed this mistake.
> > >
> > > **"non-trainable rmsnorm "scales" seem non-standard to me, have you done ablations by yourself?"**
> > >
> > > We have now added this in Figure 20 with brief explanation in Section C.1. As can be seen in the plot, this change makes a very significant difference to the transfer behavior, aligning the optimal LRs and removing the steep "cliffs" (though with fewer parameters now, the loss increases slightly). Of the two "transfer stability fixes" made in the third column of Figure 19, this is more impactful than the weight decay change.
> > >
> > > We agree that this is non-standard, though we note that it has been used for large-scale training: the OLMo model (https://arxiv.org/abs/2402.00838) adopts non-trainable RMSNorm, citing "safety" (stability?) and throughput improvements, as well as TransNormerLLM (https://arxiv.org/abs/2307.14995).
> > >
> > > ---
> > >
> > > **Summary**
> > >
> > > We are grateful to the reviewer for their thorough reading of our paper and for providing a range of feedback that we've used to improve the paper. We hope that our responses, additional experiments and modifications to the paper have been able to satisfy the questions and concerns raised. If so, we kindly ask if you’d consider increasing your review score?

---

### Official Review · Reviewer_gFHM · 2024-11-03

**Soundness:** 3
**Presentation:** 3
**Contribution:** 4
**Rating:** 8
**Confidence:** 4

**Summary:**

The authors present a combination of the maximal update parametrization (muP) and unit scaling, coined u-muP. It brings together the two main ideas of 1) hyperparameter transfer and 2) the ideal principle of unit variance of activations, weights, and gradients. The authors implement this idea with decoder language models, showing both the transfer of parameters, the performance, as well as scaling to 7B models and FP8.

**Strengths:**

I am very positive about this paper -- I think it is both a valuable contribution and an important direction for future work, as it is both practically and theoretically motivated and I agree with the concept of unit scale. I also appreciate the demonstration of failed HP transfer of muP for typical Llama-like models, which I have experienced myself. The experiments are very broad and consider not only HP transfer, but dependence between parameters, numerical properties during training, FP8 and 7B scale with downstream evaluations. Certainly, the 7B experiments are most convincing.

Some side note: As the authors note themselves (so I do not see it as a weakness, but future work), unit scaling does not give guarantees for the behavior during training, so I would be particularly interested to see this method combined with models like the outlier protected block (He et al., NeurIPS 2024 https://arxiv.org/pdf/2405.19279), investigating the outliers during training, for which this model might enable even better/easier FP8 training.

**Weaknesses:**

While I am an advocate for the paper, I want to raise some points/irregularities that came to my mind while reading and I think would need to be addressed, both to improve the work, its insights or my score. They concern, in particular, the experimental setups:

- Why not use a larger dataset for the HP transfer experiments? I understand it is to compare to the setup of Yang et al., but I am asking because it is really small compared to modern training settings. For instance, in Fig. 4, 34k steps imply ~4 epochs over the dataset? Similarly, a warmup of 2000 steps is relatively long compared to the overall steps? In comparison, the large scale training only used 500 warmup steps.
- When sweeping the LR, is the final LR always changed to 10% of the chosen rate? The final LR should either be swept independently or kept fixed to a low enough value for a proper LR cooldown, otherwise this can skew results.
- Comparison to SP, in particular 7B: If understand correctly, you use the exact same model for SP and u-muP. Does this mean you also use the tweaks (non-trainable RMSNorm, independent WD) for the SP model? Since the LR setup for Llama was chosen without those changes enabled, I think a fair comparison would be to use the original model, in particular with coupled weight decay.

I particularly think the point on the comparison to SP is very important — the main focus of the paper is on a comparison to muP, but there is the simple reason of being more convincing to adopt the method because of its performance and not just its elegance (e.g. for practitioners, many of which haven’t adopted muP yet either).

Relation to prior work: I think it would be important to add references and discussions to the field of signal propagation in neural networks, e.g. Noci et al. https://arxiv.org/pdf/2206.03126 and the many references within their related work.

## Update
I am very happy with the rebuttal and the authors' response, hence I have increased my score to 8.

**Questions:**

I have raised questions/weaknesses in the section above, which I would be happy to discuss and eventually raise my score. Beyond, I am curious what the authors think about HP transfer to larger scale and longer training. This connects to the first point above. For instance, there seems to be a slight shift of the optimal LR to the right when growing batch sizes (Fig. 4 middle). This could be problematic for planning large scale runs (where batch sizes have become enormous). Similarly, do you have concerns about transferring to much longer training lengths (e.g. more than 1M steps for Llama 3, unfeasible for LR sweeps). To be clear: I do not expect the authors to have a solution for this, I am just curious about their thoughts.

---

> ### Author Response · Authors · 2024-11-15
>
> Thanks very much for your feedback, we really appreciate the detail you're provided to make the paper better. We have a small clarifying question we wanted to ask before we write our response:
>
> You mention that a fixed LR decay percentage can skew results - this wasn't something we were aware of, but in exploring this question we came across Figures 21 (right) & 22 in [Hägele et al. (2024) ](https://arxiv.org/abs/2405.18392) which support your suggestion. We were wondering if these were the results you had in mind, or if there are others? We'll attempt to evaluate our method in light of these results if time permits.

---

> > ### Comment · Reviewer_gFHM · 2024-11-15
> >
> > Thanks a lot for your reply and asking a clarifying question! Yes, that is an illustration of what I meant. Essentially, for a large LR, the 10x final LR would also be relatively large, and therefore the final loss worse than if it were decayed more; this could change the optimal LR (e.g. for the transfer). I think it's a pretty intuitive argument and I've mostly seen it in practice and not many papers. (Though there is actually another active submission that looks at exactly this: https://openreview.net/forum?id=hrOlBgHsMI -- I'm neither an author or a reviewer there, just happened to see it right before you posted your reply :D)

---

> ### Author Response · Authors · 2024-11-23
>
> Thank you for your encouraging and constructive review. We respond as follows:
>
> **"Why not use a larger dataset for the HP transfer experiments? ... in Fig. 4, 34k steps imply ~4 epochs over the dataset"**
>
> Our use of a smaller dataset reflects the fact that our training runs do not use a large number of tokens, in order to facilitate running many full-training sweeps for the project and to keep costs down. Ideally we would train for much longer (e.g. compute-optimally), but we felt that 1 epoch of WikiText-103 was the best compromise we could viably make; still a 50x increase over the size of the original mup dataset.
>
> You rightly observe that for our batch size and steps experiments (Fig. 4) this means we use up to 4 epochs. Based on your concern (which we share) that this repeat-data regime might impact our results, we have run an additional experiment to test this, which we've added to the paper in Figure 7. This re-runs the batch size and steps experiments on the larger SlimPajama dataset, meaning no data is repeated. The change in dataset modifies the plot slightly, but the transfer properties (i.e. shape of basins) appear very similar for both, suggesting that our original results are valid.
>
> **"warmup of 2000 steps is relatively long compared to the overall steps"**
>
> We agree that this is a more significant proportion than is typically used. We did so out of caution, as we use fewer tokens-per-batch here compared with our larger model and we were concerned that in this setting using only 500 steps might not be sufficient (as best we could tell, there is no consensus on what factors should determine optimal warmup duration).
>
> We ought to have included an ablation in the original paper to validate that assumption; we have now rectified this. Figure 8 (left) shows the effect of using 500 steps of warmup versus the original 2000 steps, for different training lengths. The main effect of shorter warmup appears to be higher loss for larger learning rates, but the optima are largely unchanged. The only exception is the 4096-step run, where the optimum shifts left and the loss improves slightly. This appears to now align the optimum better with the other training durations, but leads to narrower basins as a result, suggesting a trade-off for this particular experiment.
>
> However, all our other experimental runs use the 8192-step configuration, which has a consistent optimum regardless of warmup duration here, suggesting that our choice of 2000 steps is appropriate. Figure 8 (right) shows the effect on LR transfer across width under the two regimes. The only significant impact of using less warmup is to narrow the basins, inducing no significant change in the optimal LR. As such, we conclude that using 2000 steps in our experimental setup is a reasonable choice.
>
> **"final LR should either be swept independently or kept fixed to a low enough value"**
>
> We thank the reviewer for highlighting this issue, which led us to valuable research on this question that we had not considered. As a result, we've run two experiments to investigate the effect of different levels of warmup (fixed, proportional and zero) on transfer. In Figure 9 (left) we compare these three different LR schemes and find that zero appears ideal (confirming the Cosine-D2Z results of https://openreview.net/forum?id=hrOlBgHsMI). We then re-run our width transfer experiment with decay-to-zero, and plot the result in Figure 9 (right), though especially for larger models the improvement is minimal (our 4k-width results are in progress).
>
> We anticipate that our many results in the paper which use the original setting of 10\% will still be of high practical value to the community, as this is the standard used across much of the literature (e.g. Llama, Pythia, OLMo, Falcon, Qwen) and hence will be the most applicable to the current generation of model-training.
>
> **"If understand correctly, you use the exact same model for SP and u-muP?"**
>
> No, the models do actually differ in the way you suggest they ought to, we just weren't clear about this in the text - i.e. we don't use the tweaks (non-trainable RMSNorm, independent WD) for the SP model. We have adapted section 4.4 to state explicitly that "for the SP baseline, we use a non-independent weight decay of 0.1 and parametric RMS norm as in LLama, which is standard practice".
>
> **"Add references and discussions to the field of signal propagation"**
>
> Thanks for bringing this to our attention. We've surveyed the literature here and added appropriate references to our Related Work section.
>
> [continued below...]

---

> > ### Author Response · Authors · 2024-11-23
> >
> > **"I would be particularly interested to see this method combined with models like the outlier protected block"**
> >
> > Adding architectural mitigations of this kind to protect against outliers was something we originally intended to include in this project. Specifically, we considered each of: the clipped softmax and gated attention from Bondarenko et al. 2023 (https://arxiv.org/abs/2306.12929), the explicit attention biases from Sun et al. 2024 (https://arxiv.org/abs/2402.17762) and the outlier protected block from He et al. which you mention (though we only became aware of this method later).
> >
> > Ultimately, we felt that including these in our method risked confounding the effect of our u-mup changes, making unclear what resulted from our method versus these additional features. However, additional experiments adding these techniques to u-mup (and our baseline) and re-running many of our analyses are of great interest to us. As you suggest, these could be key to providing better guarantees about being well-scaled over the course of training, and make quantization easier/stronger. We have added a paragraph to our Related Work section making the argument for use of u-mup with the methods above and encouraging this line of research.
> >
> > **"I am curious what the authors think about HP transfer to larger scale and longer training"**
> >
> > We share the reviewer's concerns about the limits of transfer along these axes. We say more on this topic in our response to CvCm's first point, where we've adjusted the paper to explicitly recommend proxy models which primarily differ in width, moderately in steps and batch size, and least in depth.
> >
> > These transfer issues we inherit from regular mup and have not sought to correct. We hope that future research will propose well-founded methods for transfer across these axes, which u-mup can then adopt.
> >
> > ---
> >
> > **Summary**
> >
> > Thanks again for giving us precise feedback about areas you see could be improved, particularly in pointing out subtleties in our experimental settings (LR decay, warmup, etc) that we'd overlooked and might affect our results. We hope that our response and several additional experiments have managed to resolve some of these points and improve the paper. If so, we ask if you’d consider increasing your score?

---

> > > ### Comment · Reviewer_gFHM · 2024-11-25
> > >
> > > I thank the authors for the detailed replies and all the new experiments and updates to the paper. I also agree with the points raised in the other replies to reviewers, e.g., the scope of the study which already includes plenty of ablations for future reference. Also, to be clear, I did not want to imply that I wanted to see combinations of the method with the other blocks you mentioned in this paper; it was just a motivation for the many potential future projects.
> > >
> > > Minor comment on reg. LR annealing: while I agree the 10% is common practice for released models, the setting there is a bit different since larger models already have lower peak LR values, so the LR will be decayed to low enough values (between 1e-5 or 1e-7). But it becomes very different when this peak LR is swept :) thanks a lot for the ablation.
> > >
> > > In summary: I am very positive about this work and think it deserves more attention, which is why I raise my score to 8. Looking forward to seeing more followups!

---

### Meta-Review · Area_Chair_J1UG · 2024-12-20

**Metareview:**

This paper proposes u-µP, a novel scheme combining Maximal Update Parametrization (µP) with Unit Scaling, to improve hyperparameter transferability and enable low-precision training for LLMs. This approach allows for efficient hyperparameter sweeps using smaller proxy models and demonstrates improved performance compared to standard µP, particularly in FP8. The reviewers noted some weaknesses, including a lack of strong empirical validation of transferability across diverse model configurations and a potentially problematic learning rate scaling for embeddings. Nevertheless, the paper's strengths lie in its novel combination of µP and Unit Scaling, its clear presentation, extensive experiments, and significant practical implications for efficient large-scale model training. Therefore I recommend acceptance.

**Additional Comments On Reviewer Discussion:**

The authors responded to questions about dataset size for hyperparameter transfer experiments by running additional tests on a larger dataset, demonstrating that their initial findings remain valid. They also addressed concerns about learning rate warmup and decay, providing new experimental results and clarifying details about their model comparison to standard parametrization. They also acknowledged the need for further investigation into embedding learning rate scaling and depth/batch size transferability, which they explained was surprising but empirically driven and warrants further research. Overall, it seems all of the reviewers main concerns were addressed.

---

### Decision · Program_Chairs · 2025-01-22

Accept (Spotlight)